# Learning Knowledge Graph-based World Models of Textual Environments

**Prithviraj Ammanabrolu**
School of Interactive Computing
Georgia Institute of Technology
raj.ammanabrolu@gatech.edu

**Mark O. Riedl**
School of Interactive Computing
Georgia Institute of Technology
riedl@cc.gatech.edu

## Abstract

World models improve a learning agent's ability to efficiently operate in interactive and situated environments. This work focuses on the task of building world models of text-based game environments. Text-based games, or interactive narratives, are reinforcement learning environments in which agents perceive and interact with the world using textual natural language. These environments contain long, multi-step puzzles or quests woven through a world that is filled with hundreds of characters, locations, and objects. Our world model learns to simultaneously: (1) predict changes in the world caused by an agent's actions when representing the world as a knowledge graph; and (2) generate the set of contextually relevant natural language actions required to operate in the world. We frame this task as a Set of Sequences generation problem by exploiting the inherent structure of knowledge graphs and actions and introduce both a transformer-based multi-task architecture and a loss function to train it. A zero-shot ablation study on never-before-seen textual worlds shows that our methodology significantly outperforms existing textual world modeling techniques as well as the importance of each of our contributions.

## 1 Introduction

World models, often in the form of probabilistic generative models, are used in conjunction with model-based reinforcement learning to improve a learning agent's ability to operate in various environments [33, 7]. They are inspired by human cognitive processes [15], with a key hypothesis being that the ability to predict how the world will change in response to one's actions will help you better plan what actions to take [12]. Evidence towards this hypothesis comes in the form of studies showing that simulating trajectories using internal learned models of the world improves sample efficiency in learning to operate in an environment [12, 30].

Text-based games, in which players perceive and interact with the world entirely through textual natural language, are environments that provide new challenges for world model approaches. Text-based games are structured as long puzzles or quests that can only be solved by navigating and interacting with potentially hundreds of locations, characters, and objects. The puzzle-like structures to the games are exacerbated by two factors. First, these environments are *partially observable*, i.e. observations of the world are incomplete. Second, the agent faces a *combinatorially-sized action space* of the order of $\mathcal{O}(10^{14})$ possible actions at every step. For these reasons model-free reinforcement learning in text-based games is extremely data inefficient.

Prior work on text-based game playing agents repeatedly demonstrated that providing agents with structured memory in the form of knowledge graphs (sets of $\langle s, r, o \rangle$ tuples such that $s$ is a subject, $r$ is a relation, and $o$ is an object) is critical in enabling them to operate in and model these worlds [3, 21, 1, 2]—aiding in both the challenges mentioned. These works all rely on *extracting* information

35th Conference on Neural Information Processing Systems (NeurIPS 2021).

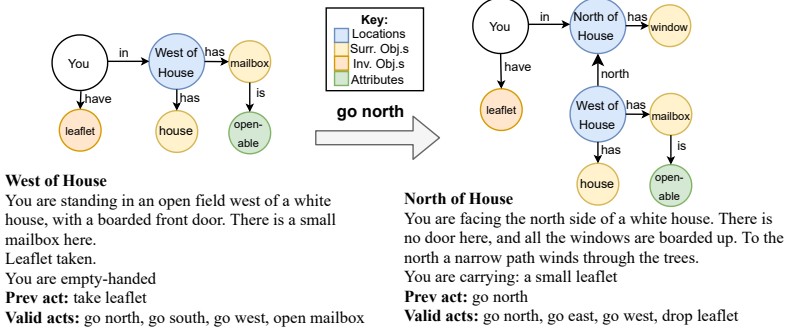

**West of House**
You are standing in an open field west of a white house, with a boarded front door. There is a small mailbox here.
Leaflet taken.
You are empty-handed
**Prev act:** take leaflet
**Valid acts:** go north, go south, go west, open mailbox

**North of House**
You are facing the north side of a white house. There is no door here, and all the windows are boarded up. To the north a narrow path winds through the trees.
You are carrying: a small leaflet
**Prev act:** go north
**Valid acts:** go north, go east, go west, drop leaflet

Figure 1: Two subsequent states in *Zork1* consisting of: textual observations, world knowledge graphs, valid actions, and actions taken.

about one's surroundings while navigating novel environments, either through rules [3, 2], question-answering [5], or transformer-based extraction [1, 21]. This *lifted* representation helps agents remember aspects of the world that become unobservable as the agent navigates the environment. However, we hypothesize that agents that rely on lifted representations of the world will benefit from more than just memorization but the ability to predict how the graph state representation will change. For example, by inferring that a locked chest is likely to contain treasure before it is actually revealed provides an agent with a form of look-ahead that will potentially enable it to bias its actions towards opening such a chest. We introduce an approach to this knowledge representation problem that effectively simplifies it by exploiting the inherent structure of knowledge graphs—framing it to be the task of *inferring the difference* in knowledge graphs between subsequent states given an action.

A consequence of the combinatorially-sized action space in text-based games is that that the set of *contextually relevant* actions—i.e. those that are most likely to affect change in the environment—are overwhelmed by the irrelevant actions. For example, it is not illegal to try to climb a tree when there are no trees present, and the game engine will just respond with feedback that nothing happens. An aspect of world modeling that has not been considered for other games is inferring which actions are valid in a particular context. We hypothesize that both the challenges mentioned are closely linked and present world models that multi-task learn to tackle both simultaneously—i.e. to answer the questions of "What actions can I perform?" and "How will the world change if I perform a particular action?".

Our work has four core contributions. (1) We first show how changes in the world can be represented in the form of differences between subsequent knowledge graph state representations. (2) We present the Worldformer, a novel multi-task transformer based architecture that learns to simultaneously generate both the *set of graph differences* and the *set of contextually relevant actions*. (3) We introduce a loss function and a training methodology that enable more effective training of the Worldformer by exploiting the fact that knowledge graphs and natural language valid actions can be represented permutation invariant Sets of Sequences—wherein the ordering of tokens within an item in the set matters but the set itself lacks ordering. (4) A zero-shot ablation study conducted on diverse set of never-before-seen text games shows the significance of each of the prior three contributions in outperforming strong existing baselines.

## 2  Related Work

We will focus on three main areas of related work: world modeling and model-based reinforcement learning, and world modeling and knowledge graphs in text games, and general (knowledge) graph construction techniques.

World modeling via model-based reinforcement learning often serves to learn transition models of an environment to allow for simulation without actually interacting with the environment [7]. Ha and Schmidhuber [12] use Variational Autoencoders (VAEs) combined with recurrent neural networks to learn compressed state representations over time of visual reinforcement learning environments [8]. This model is then used to simulate an environment and learn a control policy in it. Other contemporary works attempt to also learn dynamics models using raw pixels in the

context of games such as Atari [26, 16], and Super Mario Bros. [11] as well as 3D simulations [16] and robotics [37, 35]. We note that in all of these works, in addition to the state space being raw pixels—the action space is fixed and orders of magnitude smaller than in text games.

In textual environments, the traditional state representations of choice have been raw text encodings via recurrent neural networks [24, 14, 13] but have since shifted towards transformer [34] and knowledge graph-based representations [3, 1]. Knowledge graphs have been shown to be aid in the challenges of: (1) knowledge representation [3, 1], enabling neuro-symbolic reasoning approaches over graph-based state representations [29]; (2) combinatorial state-action spaces [2, 1]; and (3) incorporating external knowledge sources for commonsense reasoning [3, 21, 22, 9]. Two of these works are perhaps closest in spirit to ours. Yao et al. [39] train a GPT-2 model [27] to decode valid actions based on human text game transcripts found online, showing that improved valid action generation ability results in better control policies. Ammanabrolu et al. [5] frame knowledge graph construction in text games as a question-answering problem where agents ask questions to identify common objects in the world and their attributes, showing that improved knowledge graph quality results in better control policies. We note that of these works handles only one or another of the sub-tasks required for world modeling in these environments.

A core aim in the field of graph representation learning is representing graphs as continuous vectors while maintaining their inherent structure [17, 38]. Approaches to this task when applied to knowledge graphs, attempt to exploit the inherent structure of knowledge graphs to create more accurate continuous vector graph representations in the form of embeddings [36]. Building on these representation learning works is the task of automated knowledge base construction attempts to create links in knowledge graphs given text [25]. Li et al. [19] approach the graph generation problem as a sequential process, first learning a generative model of the graph and then iteratively adding nodes and edges to it using the learned model—this work does not consider cases where the graphs are conditioned on text, however. These works focus solely on graph construction and do not include the inherent interactive action-based components featured in world modeling and text games.

## 3 Background

**Dataset.** We use the JerichoWorld Dataset [4].[1] It contains 24,198 mappings between rich natural language observations and: (1) knowledge graphs in the form of a set of tuples $\langle s, r, o \rangle$ (such that $s$ is a subject, $r$ is a relation, and $o$ is an object) that reflect the world state in the form of a map; (2) a set of natural language actions that are guaranteed to cause a change in that particular world state. An example of the mapping between rich natural language observations and structured knowledge is illustrated in Figure 1. The training data is collected across 27 text games in multiple genres and contains a further 7,836 heldout instances over 9 additional games in the test set.

Each instance of the dataset takes the form of a tuple of the form $\langle S_t, A, S_{t+1}, R \rangle$ where $S_t$ and $S_{t+1}$ are two subsequent states with $A$ being the action used to transition between states and $R$ the observed reward. As mentioned earlier, each of the states in the tuple contain information regarding the observation $O_t \in S_t$, ground truth knowledge graph $G_t \in S_t$, and valid actions for that state $V_t \in S_t$. This data was collected by oracle agents, i.e. agents that can perfectly solve a game, exploring using a mix of an oracle and random policy to ensure high coverage of a game's state space. A full sample is found in Appendix A.1.

**Tasks.** Given this dataset, we focus on two tasks within it as formally defined by JerichoWorld. As mentioned in Section 1, a successful world model will be able to accomplish both of these tasks.

1. **Knowledge Graph Generation:** this task involves predicting the graph at time step $t + 1$ : $G_{t+1} \in S_{t+1}$ given the textual observations, valid actions, and graph at time step $t$ : $O_t, V_t, G_t \in S_t$, and action $A$ for all samples in the dataset.

2. **Valid Action Generation:** this task is formally defined as predicting the set of sequences of valid actions at time step $t + 1$ : $V_{t+1} \in S_{t+1}$ given the textual observations, valid actions, and graph at time step $t$ : $O_t, V_t, G_t \in S_t$, and action $A$ for all samples in the dataset.

---

[1]https://github.com/JerichoWorld/JerichoWorld

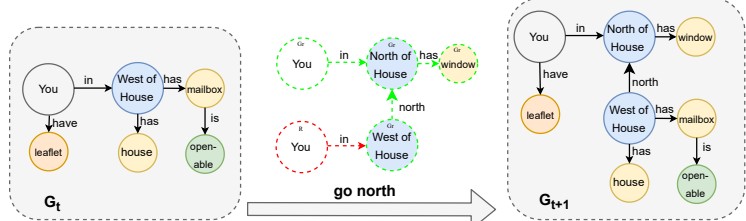

Figure 2: The transformation between subsequent world knowledge graphs $G_t$ and $G_{t+1}$ based on the states in Figure 1. The green (Gr) outlined portions in the center are additions to $G_t$ to get $G_{t+1}$ (i.e. $G_{t+1} - G_t$) and the red (R) portions similarly represent deletions to $G_t$ (i.e. $G_t - G_{t+1}$).

## 4 The Worldformer

This section describes the core methodological contributions of our work in creating world models for text games. We first show how knowledge graph generation can be simplified to predicting the graph difference between agent steps. We then describe the Worldformer, a transformer-based architecture, and end-to-end training method—including an objective function—that treats both of the world modeling tasks as a Set of Sequences generation problem.

### 4.1 Knowledge Graph Difference Generation

Figure 2 describes the gist of our simplification of the knowledge graph generation problem. Recall that knowledge graphs are directed graphs that are stored the form of a *set of tuples* as $\langle s, r, o \rangle$ such that $s$ is a subject, $r$ is a relation, and $o$ is an object. Let the knowledge graphs representing the world state at two subsequent steps be $G_t$ and $G_{t+1}$. At every step, tuples are either added or deleted from the graph $G_t$ to update the belief state about the world and turn it into graph $G_{t+1}$. Using this observation, we can simplify the knowledge graph generation problem. Instead of predicting $G_{t+1}$ given $G_t$ and prior context, we can instead predict the *differences* between the two graphs.

In Figure 2, between steps $t$ and $t + 1$, we see that $G_{t+1} - G_t$ is the set of tuples that are added to $G_t$ and $G_t - G_{t+1}$ the set of tuples are are deleted from $G_t$. Together they make up the graph differences. Here, we make a second key observation that allows for yet further simplification of the problem. This observation is based on generally applicable properties of such worlds: (1) locations are fixed and unique, i.e. the positions of locations with respect to each other does not change; (2) objects and characters can only be in one location at a time; and (3) contradicting object attributes can be identified using a lexical dictionary such as WordNet [20], e.g. an object cannot be both open and closed at the same time. These properties let us uniquely identify the triples to be deleted from the graph $G_t - G_{t+1}$ given triples to be added to the graph $G_{t+1} - G_t$. Additional implementation details can be found in Appendix A.2.

Taken together, the Knowledge Graph Generation task can be cast as follows: predict the nodes to be added to the graph $G_t$ at time step $t : G_{t+1} - G_t$ (a much smaller set than $G_{t+1}$ by itself) to transform it into graph $G_{t+1}$ given the textual observations, valid actions, and graph at time step $t : O_t, V_t, G_t \in S_t$, and action $A$ for all samples in the dataset.

### 4.2 Multi-task Architecture

The Worldformer is a multi-task world model that simultaneously learns to perform both knowledge graph and valid action generation. It is built on the hypothesis that each of these tasks contains information crucial to the other—the valid actions that can be executed at any timestep are entirely dependent on the current state and vice versa the state knowledge graph updates on the basis of the previously executed action.

Figure 3 describes the architecture of the Worldformer. The inputs to the architecture are textual observations, valid actions, and graph at time step $t : O_t, V_t, G_t \in S_t$. $O_t$ and $V_t$ are encoded through a bidirectional text encoder into $\mathbf{O_t}$. In our work, we used an architecture similar to BERT [10] with the original pre-trained weights that are then fine-tuned using a masked language model (MLM) loss on observations taken from the training data. $\mathbf{O_t}$ is the output of the final hidden layer. The

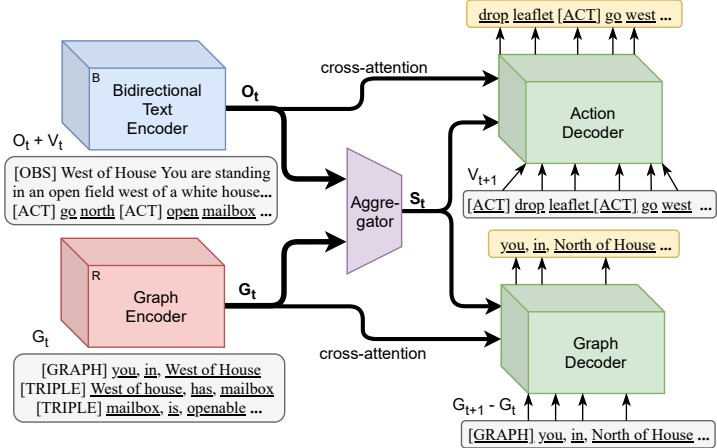

Figure 3: The Worldformer architecture. The text encoder (B) and graph encoder (R) have similar architecture but different pre-training strategies. Both the decoders are not pre-trained and have identical architectures. Solid black lines indicate gradient flow.

graph encoder receives $G_t$ and encodes it into $\mathbf{G_t}$. It is also similar to BERT, but is pre-trained on knowledge graphs found in the training data using a MLM loss with a phrase-level masking scheme where whole components of a $\langle s, r, o \rangle$ graph triple (individual underlined portions in Figure 3) are masked at once. Again, $\mathbf{G_t}$ is the output of the final hidden layer.

$\mathbf{O_t}$ and $\mathbf{G_t}$ are passed into a representation aggregator which then sends the combined encoded state representation $\mathbf{S_t}$ to one of two autoregressive decoders that have the same general internal architecture as GPT-2 [27]. During training, the first decoder is conditioned on $\mathbf{S_t}$ directly and $\mathbf{O_t}$ through cross-attention and takes in the valid actions of the next state $V_{t+1}$ as input, learning to predict the same input sequence shifted to the right as sequence-to-sequence models do. Similarly, the second decoder is conditioned on $\mathbf{S_t}$ directly and $\mathbf{G_t}$ through cross-attention and takes in the knowledge graph of the next state $G_{t+1}$ as input.

### 4.3 Set of Sequences Generation and Training

We observe that both the knowledge graph difference $G_{t+1} - G_t$ and the valid actions $V_{t+1}$ are both *Sets of Sequences* where the ordering of the sequence of tokens within an action or a graph triple matters but the ordering of all the actions and triples does not. Standard autoregressive decoding used in sequence-to-sequence (Seq2Seq) models [32] does not account for such permutation invariance. We frame the graph and action prediction tasks as a generation of a Set of Sequences (SOS) problem—expanding on the simple set prediction problem definition proposed by works such as Deep Sets [40] to account for the specific structure of Sets of Sequences. This problem structure is used to then formulate a training methodology that lets autoregressive decoders better account for the SOS structure.

For both of the decoders in Figure 3, we are given a target sequence $Y = \{y_1, ..., y_M\}$ and some input context via the encoders $X$. Standard autoregressive techniques factor the distribution over the target sequence into a chain of conditional probabilities with a causal left to right structure.

$$P(Y|X;\theta) = \prod_{i=1}^{M+1} p(y_i|y_{0:i-1}, X; \theta) \tag{1}$$

Where $\theta$ represents the overall network parameters. This can then be used to formulate a maximum likelihood training loss with cross-entropy at every step.

$$\mathcal{L}_{seq} = \log P(Y|X;\theta) = \sum_{i=1}^{M+1} \log p(y_i|y_{0:i-1}, X; \theta) \tag{2}$$

In our setting, we can group elements in $Y$ into its set of sequences form

$$Y_{\text{sos}} = \{y'_1...y'_{M'}\}, y'_i \in V_{t+1} \text{ or } y'_i \in G_{t+1} - G_t, M' \leq M$$

$$\text{where } y'_i = \{y_k...y_{k+l}\}, \sum_j \text{len}(y'_j) = M \quad (3)$$

Via the decoders, we seek to learn a transformation from $\mathbf{S_t} \in \mathbb{R}^d$ (the input $d$-dimensional state representation vector) and $Y_{\text{sos}} \in \mathcal{Y}$ (decoder inputs in the space of all possible decoder inputs $\mathcal{Y}$) that map to the permutation invariant target set of sequences $Y_{\text{sos}}$. This function can then be defined as $f : \mathbb{R}^d \cup 2^{\mathcal{Y}} \rightarrow 2^{\mathcal{Y}}$ as the permutation invariance of part of the domain and range of this function makes it the power set of $\mathcal{Y}$.

Combining this definition of permutation invariant functions with Eq. 2, 3, we can factorize the distribution over the output Set of Sequences as the following chain of probabilities:

$$P(Y_{\text{sos}}|X; \theta) = \prod_{i=1}^{M+1} p(y'_i|X; \theta) \quad (4a)$$

$$p(y'_i|X; \theta) = \prod_{k=l}^{l+n} p(y_k|y_{l:k-1}, X; \theta) \quad (4b)$$

$$\text{where } l = \sum_{j<i} \text{len}(y'_j), \ n = \text{len}(y'_i)$$

With the key intuition here being that Eq. 4a factorizes the distribution such that each element of $Y_{\text{sos}}$ is independent of other elements in the set, but tokens of an element $y'_i$ in the set are conditioned on preceding tokens within the element (Eq. 4b).

This in turn gives us a maximum likelihood Set of Sequences loss that can be used to train a model to output a Set of Sequences.

$$\mathcal{L}_{\text{sos}} = \log P(Y_{\text{sos}}|X; \theta) = \sum_{i=1}^{M+1} \log p(y'_i|X; \theta)$$

$$= \sum_{i=1}^{M+1} \sum_{k=l}^{l+n} \log p(y_k|y_{l:k-1}, X; \theta) \quad (5a)$$

$$\text{where } l = \sum_{j<i} \text{len}(y'_j), \ n = \text{len}(y'_i) \quad (5b)$$

In our formulation, we have observation sequences at timestep $t : O_t, V_t$ encoded into $\mathbf{O_t}$, graph $G_t$ encoded into $\mathbf{G_t}$, and all of them combined into $\mathbf{S_t}$, with the output Sets of Sequences at timestep $t + 1$ being the graph difference $G_{t+1} - G_t$ and valid actions $V_{t+1}$. Across the two decoders, this gives us a combined loss:

$$\mathcal{L}_{\text{world}} = \log P(G_{t+1} - G_t|\mathbf{S_t}, \mathbf{G_t}; \theta) + \log P(V_{t+1}|\mathbf{S_t}, \mathbf{O_t}; \theta) \quad (6)$$

This loss is used to multi-task train the Worldformer simultaneously across the two tasks.

## 5 Evaluation

We evaluate the Worldformer by comparing it on both of the tasks across 9 never-before-seen testing games against strong baselines. We further present ablation studies in each task to determine the relative importance of each of the techniques presented in the previous section.

**Metrics.** Across both the tasks, we use the same metrics as defined in JerichoWorld [4]. For knowledge graph generation, we report two types of metrics (Exact Match or EM and F1) operating on two different levels—at a *graph* tuple level and another at a *token* level. The graph level metrics are based on matching the set of ⟨*subject, relation, object*⟩ triples within the graph, all three tokens in a particular triple must match a triple within the ground truth graph to count as a true positive. The

| Expt. | Metrics | Game | zork1 | lib. | det. | bal. | pent. | ztuu | ludi. | deep. | temp. | overall |
|---|---|---|---|---|---|---|---|---|---|---|---|---|
| | rics | Size | 886 | 654 | 434 | 990 | 276 | 462 | 2210 | 630 | 1294 | 7836 |
| **Knowledge Graph Prediction** | | | | | | | | | | | | |
| Rules | Gr. | EM | 3.72 | 7.61 | 1.39 | 9.17 | 6.44 | 4.94 | 5.10 | 0.49 | 2.48 | 4.70 |
| | | F1 | 4.46 | 12.87 | 4.55 | 11.90 | 10.22 | 10.06 | | 0.64 | 3.36 | 7.25 |
| | Tok. | EM | 6.08 | 10.33 | 7.51 | 32.53 | 16.48 | 14.40 | 14.47 | 3.34 | 7.42 | 13.08 |
| | | F1 | 8.42 | 26.74 | 10.23 | 36.09 | 23.36 | 21.74 | 18.48 | 3.86 | 9.44 | 17.50 |
| Q*BERT (Question Answering) | Gr. | EM | 24.56 | 29.14 | 34.45 | 41.22 | 28.96 | 22.17 | 41.44 | 4.42 | 36.84 | 32.79 |
| | | F1 | 24.88 | 31.46 | 36.23 | 41.85 | 30.12 | 26.26 | 46.74 | 4.66 | 39.86 | 35.48 |
| | Tok. | EM | 43.93 | 49.78 | 60.28 | 85.81 | 65.02 | 49.44 | 57.58 | 9.31 | 48.98 | **53.58**† |
| | | F1 | 48.31 | 52.76 | 63.21 | 86.18 | 69.54 | 49.82 | 60.95 | 9.84 | 49.17 | **55.74**† |
| Seq2Seq | Gr. | EM | 12.44 | 18.42 | 26.86 | 8.19 | 22.18 | 16.89 | 12.94 | 8.38 | 16.48 | 14.29 |
| | | F1 | 12.96 | 18.89 | 29.48 | 9.04 | 23.54 | 16.89 | 14.18 | 10.47 | 18.52 | 15.54 |
| | Tok. | EM | 18.01 | 20.26 | 35.86 | 17.60 | 25.48 | 17.19 | 14.8 | 13.25 | 22.48 | 18.80 |
| | | F1 | 21.12 | 20.84 | 35.86 | 18.86 | 27.72 | 17.87 | 15.42 | 13.25 | 24.34 | 19.96 |
| GATA-W | Gr. | EM | 22.30 | 24.72 | 21.72 | 23.68 | 22.81 | 27.00 | 24.55 | 23.76 | 24.52 | 24.06 |
| | | F1 | 25.34 | 26.47 | 22.14 | 26.54 | 27.63 | 27.00 | 24.55 | 23.76 | 24.92 | 25.19 |
| | Tok. | EM | 33.09 | 33.88 | 25.64 | 34.64 | 37.71 | 35.81 | 35.94 | 32.48 | 40.89 | 35.31 |
| | | F1 | 33.93 | 34.86 | 25.80 | 38.68 | 39.59 | 38.88 | 37.16 | 32.48 | 43.97 | 37.10 |
| Worldformer | Gr. | EM | 21.62 | 34.39 | 41.05 | 50.41 | 30.00 | 41.56 | 40.10 | 41.87 | 42.43 | **39.15*** |
| | | F1 | 24.44 | 34.39 | 44.53 | 52.43 | 34.30 | 42.20 | 41.65 | 42.74 | 45.17 | **41.06*** |
| | Tok. | EM | 42.88 | 41.98 | 54.39 | 62.22 | 49.00 | 50.80 | 51.29 | 50.04 | 53.81 | 51.32 |
| | | F1 | 48.12 | 41.98 | 59.13 | 62.22 | 49.00 | 52.24 | 51.29 | 50.04 | 54.96 | 52.45 |
| **Valid Action Prediction** | | | | | | | | | | | | |
| Seq2Seq | Act | EM | 16.65 | 15.13 | 18.19 | 16.19 | 23.39 | 14.75 | 20.10 | 14.71 | 20.34 | 18.10 |
| | | F1 | 17.85 | 16.88 | 21.12 | 18.23 | 25.87 | 15.13 | 20.86 | 14.86 | 22.14 | 19.44 |
| CALM | Act | EM | 18.67 | 11.18 | 17.37 | 10.04 | 13.77 | 11.29 | 15.49 | 10.31 | 13.13 | 13.79 |
| | | F1 | 18.90 | 25.49 | 34.42 | 12.16 | 34.40 | 9.95 | 20.94 | 7.84 | 18.57 | 19.11 |
| Worldformer | Act | EM | 23.08 | 22.55 | 20.97 | 29.08 | 27.05 | 20.71 | 21.36 | 24.04 | 22.80 | **23.22*** |
| | | F1 | 23.50 | 26.52 | 25.28 | 32.89 | 31.32 | 23.66 | 22.27 | 26.12 | 25.66 | **25.54*** |

Table 1: Results across both the tasks for the models specified. Overall indicates a size weighted average. All experiments were performed over three random seeds, with standard deviations not exceeding $\pm 3.2$ in any of the overall categories for KG prediction and $\pm 1.2$ for valid action prediction. Bolded results indicate highest overall scores. Asterisk ($*$) indicates when the top result is significantly higher ($p < 0.05$ with an ANOVA test followed by a post-hoc pair-wise Tukey test) over all alternatives. † indicates this result is *not* significantly higher than Worldformer.

token level metrics operate on measuring unigram overlap in the graphs, any relations or entities in the predicted tokens that match the ground truth count towards a true positive.

For valid action generation, we adapt the graph level Exact Match (EM) and F1 metrics as described in the previous task to actions. In other words, positive EM or F1 happens only when all tokens in a predicted valid action match one in the gold standard set. In all cases, EM checks for accuracy or direct overlap between the predictions and ground truth, while F1 is a harmonic mean of predicted precision and recall.

## 5.1 Knowledge Graph Generation

We compare the Worldformer to 4 baselines taken from contemporary knowledge graph-based world modeling approaches in text games. All sequence models use a fixed graph vocabulary of size 7002 that contains all unique relations and entities at train and test times. Additional details and hyperparameters for the models are found in Appendix A.2.

**Rules.** Following Ammanabrolu and Hausknecht [2], we extract graph information from the observation using information extraction tools such as OpenIE [6] in addition to some hand-authored rules to account for the irregularities of text games.

**Question-Answering.** (QA) This baseline comes from the Q*BERT agent described in [5]. It is trained on both the SQuAD 2.0 [28] the Jericho-QA text game question answering dataset [5] on the same set of training games as found in Worldformer. It uses the ALBERT [18] variant of the BERT [10] natural language transformer to answer questions and populate the knowledge graph via a few hand-authored rules from the answers. Examples of questions asked include: "What is my current location?", "What objects are around me?".

**Seq2Seq.** This single-task model is provided as a baseline by JerichoWorld and performs sequence learning by encoding the observation and graph with a single bidirectional BERT-based encoder

and using an autoregressive GPT-2-based decoder to decode the next graph. It is trained using the standard Seq2Seq cross-entropy loss (Eq. 2).

**GATA-World.** We adapt the Graph-Aided Transformer Agent [1] to our task. It consists of the same encoder structure as the Worldformer but contains one decoder that performs single-task Seq2Seq learning to decode both the set of tuples that must be added as well as deleted from the graph in the form of: ⟨*add, node1, node2, relation*⟩ or ⟨*del, node1, node2, relation*⟩. This is equivalent to predicting $(G_{t+1} - G_t) \cup (G_t - G_{t+1})$. It is trained with the Seq2Seq cross-entropy loss (Eq. 2).

Table 1 describes the results in this task over all the games. We see that on the graph level metrics, the Worldformer performs significantly better than all other other baselines. On the token level metrics, the Worldformer and QA method are comparable—the difference between these two methods are statistically non-significant ($p = 0.18$) with each other but both significantly ($p < 0.05$) higher than all others. The QA method, and other extractive methods, highlight portions of the input observation that form the graph and are particularly well suited for the token level metrics. The JerichoWorld developers note that these approaches are prone to over-extraction, i.e. extracting more text than is strictly relevant from the input observation aiding token level overlap but resulting in a sharp drop in terms of the graph level metrics [4]. Additional failure modes of such extraction based approaches occur when the text descriptions are incomplete or hidden—e.g. the contents of a chest are revealed through the textual observation only when it is opened by a player. The Worldformer is able to make a informed guess as to the contents of the chest due to its training, providing a form of look ahead that the Rules and QA systems cannot.

Table 3 present the results of an ablation study testing the relative importance of the three main components of the Worldformer: graph difference prediction, multi-task training, and the SOS loss. We note that a model without any of these components is equivalent to the Seq2Seq approach described previously. We see significant drops in performance, particularly on the graph level metrics, when any single one of these components are removed. This indicates that all three components are necessary for the Worldformer to achieve state-of-the-art performance.

| Ablation | | | Graph | | Token | |
|---|---|---|---|---|---|---|
| Graph Diff | Multi Task | SOS Loss | EM | F1 | EM | F1 |
| | | | 14.29 | 15.54 | 18.80 | 19.96 |
| | ✓ | ✓ | 29.29 | 31.41 | 39.99 | 41.02 |
| ✓ | | ✓ | 32.60 | 34.65 | 42.74 | 44.35 |
| ✓ | ✓ | | 35.94 | 36.17 | 48.82 | 50.18 |
| ✓ | ✓ | ✓ | **39.15** | **41.06** | **51.32** | **54.45** |

Table 2: Worldformer ablations to test the impact of its three main components for KG prediction. All results are size weighted averages over all test games over three random seeds, with standard deviations not exceeding $\pm 3.2$ in any category.

In particular, we note that the largest performance drop was when Worldformer did not use the graph difference simplification. In this case, the KG prediction task is simplified to predicting only; the length of the set of sequences $G_{t+1} - G_t$ is much smaller than $G_{t+1}$. There are on average 3.42 triples or 10.42 tokens per state across the JerichoWorld test dataset for $G_{t+1} - G_t$ but a mean of 8.71 triples or 26.13 tokens per state for $G_{t+1}$. This also explains the increased performance of the GATA-W over the baseline Seq2Seq agent—this agent only needs to predict on average 5.04 rules or 20.16 tokens across the testing games. Predicting a smaller number of triples and tokens per state makes the problem relatively more tractable for world modeling agents.

## 5.2 Valid Action Generation

Similarly to the other task, we compare the Worldformer to an existing baseline for valid action prediction. All models use a fixed vocabulary of size 11,056 at train and test times.

**Seq2Seq.** This single-task model is provided as a baseline by JerichoWorld and is identical to the Seq2Seq model described in the previous task but is single-task trained to predict valid actions.

**CALM.** A complementary dataset of observation-action pairs created by humans on the ClubFloyd online Interactive Narrative forum[2] appears in both Ammanabrolu and Hausknecht [2] and Yao et al. [39] with the latter using it to tune a GPT-2 model for valid action prediction using a GPT-2 based Seq2Seq valid action model dubbed CALM.[3] This model takes in $O_t, A, O_{t+1}$ and targets $V_{t+1}$.

In Table 1, we see that the Worldformer significantly outperforms the Seq2Seq baseline on all the games and CALM overall. Each valid action in a text game requires at most 5 tokens. This combined with an average of 10.30 valid actions per test state means that for every state we would need to generate about 52 tokens. Yet further, the vocabulary size for actions is $11,056$, larger than the graph

---

[2]http://www.allthingsjacq.com/interactive_fiction.html
[3]https://github.com/princeton-nlp/calm-textgame

vocabulary of $7,002$. This increase in task difficulty explains the relative decrease in the magnitude of performance metrics between KG and valid action prediction tasks. Both the Seq2Seq model and CALM—which is trained on a different dataset—are comparable on F1 scores but Seq2Seq is better overall for exact matches. CALM also has relatively higher variance in performance across the test games than the other two methods—e.g. on some games such as *zork1* and *detective* it outperforms the Seq2Seq and is not too far off the Worldformer especially in terms of F1 score. This would appear to indicate that the Club Floyd dataset of text game transcripts that CALM was trained on is better suited for transfer to certain games than others due to differences in training set genre similarities.

Table 3 presents an ablation study that tests the two main components of the Worldformer for this task: multi-task learning, and SOS loss. As with the KG prediction task, we observe significant drops in performance when either of these components are taken away—suggesting that they are relatively critical components. The JerichoWorld developers note that there is a correlation between performance of the baseline Seq2Seq model to the average number of valid actions for the testing game (see Appendix Table 4). They attribute this to label imbalance in the dataset, stating that the model likely learns a common set of actions found across all games such as navigation actions before learning more fine-grained actions. E.g. *ztuu, deephome,* and *balances* have a high number of gold standard average valid actions while *pentari, ludicorp, detective,* and *temple* which have a low number of average valid actions. While the latter set of games have generally higher performance on both the Seq2Seq and Worldformer models, the gap is significantly less pronounced with the Worldformer. We hypothesize that this is due to the multi-task training of the Worldformer—encoder representations now contain enough information regarding the next knowledge graph to alleviates the label imbalance of the actions and enable prediction of more fine-grained actions.

| Ablation | | Act | |
|---|---|---|---|
| Multi Task | SOS Loss | EM | F1 |
| | | 18.10 | 19.44 |
| | ✓ | 20.78 | 22.42 |
| ✓ | | 20.12 | 21.28 |
| ✓ | ✓ | **23.22** | **25.54** |

Table 3: Worldformer ablations to test the impact of its two main components for action prediction. All results are size weighted averages over all test games over three random seeds, with standard deviations not exceeding $\pm 1.2$.

## 6    Conclusions

We presented the Worldformer, a state-of-the-art world model for text games that: maps worlds by predicting the difference in knowledge graphs between subsequent states, multi-task learns to map a world and act in it simultaneously, and frames all of these tasks as a Set of Sequences generation problem. Its state-of-the-art performance and an ablation study have three potential implications: (1) the simplification of the knowledge representation problem into that of predicting knowledge graph differences between subsequent states is a critical step in making the problem more tractable; (2) performance improvements due to multi-task training imply that acting in and mapping these worlds is inherent a highly correlated problem and benefits from being solved jointly; and (3) the performance boosts due to the SOS loss suggest that accounting for this property of graphs and actions enables more effective training than if we were to treat them as simple sequences.

## 7    Broader Impacts

We view text-games as an platform on which to teach agents how to communicate effectively using natural language, to plan via sequential decision making in situations that may not be anticipated. We seek to enable agents to more efficiently model such worlds, helping them produce more contextually relevant language in such situation. As stated in JerichoWorld—and further verified by us—data is collected from games containing situations of non-normative language usage—describing situations that fictional characters may engage in that are potentially inappropriate, and on occasion impossible, for the real world such as running a troll through with a sword. Instances of such scenarios are mitigated by careful curation of the games that the data is collected by both the authors of Jericho [13] and JerichoWorld [4]. This is based on manual vetting and (existing) crowd-sourced reviews on the popular interactive narrative forum IFDB (https://ifdb.org/).

Broadly speaking, the most relevant downstream task for this work is model-based reinforcement learning. It is applicable to many sequential tasks, some of which cannot be anticipated. World modeling for text environments is more suited for domains in which change in the world is affected via language, which mitigates physical risks—downstream lines of work are not directly relevant to robotics—but not cognitive and emotional risks, as any system capable of generating natural language is capable of accidental or intentional non-normative and biased language use [23, 31].

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
