# A   Appendix

## A.1   Dataset

An example of a single state from a $\langle S_t, A, S_{t+1}, R \rangle$ tuple taken from the JerichoWorld dataset. This dataset is licensed using the MIT license.

```
Game: ztuu
Location: Cultural Complex This imposing ante-room, the center of what was apparently the cultural center
    of the GUE, is adorned in the ghastly style of the GUE's "Grotesque Period." With leering gargoyles,
    cartoonish friezes depicting long-forgotten scenes of GUE history, and primitive statuary of pointy-
    headed personages unknown (perhaps very, very distant progenitors of the Flatheads), the place would
    have been best left undiscovered. North of here, a large hallway passes under the roughly hewn
    inscription "Convention Center." To the east, under a fifty-story triumphal arch, a passageway the
    size of a large city boulevard opens into the Royal Theater. A relatively small and unobtrusive sign
    (perhaps ten feet high) stands nearby. South, a smaller and more dignified (i.e. post-Dimwit) path
    leads into what is billed as the "Hall of Science." You can see a pair of razor-like gloves here.
Observation: You put on the razor-like gloves.
Inventory:
    You are carrying:
      a brass lantern (providing light)
      a pair of glasses
      four candy bars:
        a ZM$100000
        a Multi-Implementeers
        a Forever Gores
        a Baby Rune
      a cheaply-made sword
Prev Act: put on gloves
Inventory Objects:
    candy: Which do you mean, the ZM$100000, the Multi Implementeers, the Forever Gores or the Baby Rune?
    Implementeers: The profiles on the wrapper of this delicacy look more like Moe, Larry, and Curly than
        those of your favorite Implementeers (presumably, Marc, Mike, and David.)
    Forever/Gores: The wrapper of this bar pictures the Milky Way, but the stars are all blood red. Kids
        love them.
    sword: This is a cheaply made sword of no antiquity whatsoever. With regard to grues or other
        underworldly denizens, your weapon is as likely to engender laughter as fear.
    rune: The label is covered with mystical runes, the meanings of which elude you.
    glasses: The owner of these glasses had an indeterminate vision problem, because the lenses have both
        been crushed underfoot. The vision problem, of course, has been solved.
    lantern: The lantern, while of the cheapest construction, appears functional enough for the moment.
        Your best hope is that it stays that way. It looks like the lamp has gone through a few cycles of
         impact revitalization.
Inventory Attributes:
    glasses: clothing
    gloves: clothing
    sword: animate, equip
    lantern: animate, equip
Surrounding Objects:
    gargoyles: Unless you are inordinately masochistic, the less time spent examining the artwork, the
        better.
    east: You see nothing special about the east wall.
    tunnel: The tunnel leads west.
    gloves: The razor like gloves would be very attractive for an axe murderer. And they're just your size.
    south: You see nothing special about the south wall.
    sign: The sign indicates today's performance, which (in honor of the festivities in the Convention
        Center) is "A Massacre on 34th Street."
Surrounding Attributes:
    gloves: clothing
    tunnel: animate
    sign: animate
Graph: [sign, in, Cultural Complex], [you, have, Forever Gores], [you, have, ZM$100000], [you, have, Baby
    Rune], [tunnel, in, Cultural Complex], [you, in, Cultural Complex], [you, have, brass lantern], [you,
    have, glasses], [decoration, in, Cultural Complex], [you, have, cheaply-made sword], [you, have,
    Multi-Implementeers], [you, have, razor-like gloves], [glasses, is, clothing], [gloves, is, clothing],
    [sword, is, animate], [tunnel, is, animate], [sign, is, animate], [lantern, is, animate], [sword, is,
    equip], [lantern, is, equip]
Valid Actions: west, turn lantern off, east, south, put multi down, put forever down, put lantern down,
    put rune down, put glasses down, put sword down, take razor off, put on glasses, examine glasses,
    lower razor, throw multi, throw lantern, put multi in glasses, north
```

## A.2   Training and Hyperparameters

All baseline models have hyperparameters taken from their respective works and from the JerichoWorld benchmarks. They are trained accordingly, with the exception of GATA-World. This model uses an architecture identical to that of the Worldformer but is trained to predict add/del

| Game | No. Samples | Input Vocab Size | Avg. Obs Token Len. | Avg. No. Valid Actions | Avg. Graph Triple Len. | Avg. Graph Diff Len. |
|---|---|---|---|---|---|---|
| **Training games** | | | | | | |
| wishbringer | 560 | 1043 | 136.54 | 10.35 | 4.00 | 2.17 |
| snacktime | 168 | 468 | 190.08 | 4.82 | 2.33 | 0.98 |
| tryst205 | 1052 | 871 | 136.24 | 14.30 | 7.81 | 4.33 |
| enter | 448 | 470 | 219.06 | 18.04 | 14.79 | 4.55 |
| omniquest | 784 | 460 | 79.96 | 21.50 | 8.02 | 3.05 |
| zork3 | 1142 | 564 | 137.68 | 12.72 | 6.59 | 2.78 |
| zork2 | 584 | 684 | 154.90 | 29.66 | 7.82 | 4.11 |
| inhumane | 1004 | 409 | 90.24 | 4.31 | 3.86 | 1.61 |
| 905 | 504 | 296 | 100.91 | 13.60 | 11.69 | 2.73 |
| loose | 8 | 1141 | 140.38 | 2.12 | 10.12 | 5.25 |
| murdac | 1914 | 251 | 80.76 | 8.67 | 4.30 | 1.85 |
| moonlit | 684 | 669 | 131.62 | 9.20 | 12.10 | 4.49 |
| dragon | 894 | 1049 | 182.79 | 13.13 | 11.64 | 5.26 |
| jewel | 1418 | 657 | 119.08 | 13.82 | 7.21 | 2.80 |
| weapon | 294 | 481 | 230.41 | 9.65 | 29.79 | 8.14 |
| karn | 2196 | 615 | 138.87 | 26.36 | 13.24 | 3.63 |
| zenon | 402 | 401 | 101.52 | 5.97 | 5.01 | 2.06 |
| acorncourt | 474 | 343 | 323.38 | 20.18 | 36.14 | 4.37 |
| ballyhoo | 2132 | 962 | 127.08 | 15.39 | 7.25 | 2.77 |
| yomomma | 884 | 619 | 129.06 | 16.11 | 3.00 | 0.58 |
| enchanter | 1714 | 722 | 133.56 | 45.27 | 14.83 | 3.98 |
| gold | 2082 | 728 | 166.96 | 25.03 | 15.76 | 6.16 |
| huntdark | 344 | 539 | 162.33 | 6.33 | 13.01 | 4.04 |
| afflicted | 574 | 762 | 165.13 | 17.34 | 2.91 | 0.72 |
| adventureland | 870 | 398 | 87.41 | 9.02 | 6.99 | 3.07 |
| reverb | 722 | 526 | 101.92 | 9.04 | 5.23 | 1.83 |
| night | 346 | 462 | 49.92 | 4.55 | 10.17 | 1.27 |
| overall train | 24198 | 11056 | 133.30 | 17.41 | 9.74 | 3.30 |
| **Testing games** | | | | | | |
| deephome | 630 | 760 | 147.33 | 15.31 | 10.20 | 5.88 |
| balances | 990 | 452 | 107.15 | 13.04 | 7.61 | 2.41 |
| ludicorp | 2210 | 503 | 88.32 | 9.27 | 9.47 | 3.66 |
| pentari | 276 | 472 | 130.34 | 3.72 | 3.46 | 2.75 |
| detective | 434 | 344 | 105.97 | 5.72 | 2.80 | 1.64 |
| ztuu | 462 | 607 | 170.89 | 18.39 | 11.97 | 4.32 |
| zork1 | 886 | 697 | 109.70 | 13.02 | 6.46 | 3.80 |
| library | 654 | 510 | 154.40 | 4.59 | 9.18 | 4.95 |
| temple | 1294 | 622 | 138.07 | 8.56 | 10.77 | 4.15 |
| overall test | 7836 | 11056 | 118.92 | 10.30 | 8.71 | 3.42 |

Table 4: Statistics for JerichoWorld especially showcasing the difference between the average number of graph triples per state, and the average graph difference length per instance.

rules as described in Section 5—i.e. it is trained single-task and has only a graph decoder and no action decoder. The hyperparameters and training methodology for this model match those of the Worldformer described below.

### A.2.1 Worldformer

Following JerichoWorld [4] models were trained until validation accuracy (picked to be a random 10% subset of the training data) did not improve for 5 epochs or 96 wall clock hours on a machine with 4 Nvidia GeForce RTX 2080 GPUs, three times with three random seeds. All models decode using beam search with a beam width of 15 at test time until the end-of-sequence tag is reached. The size of the decoding vocabulary for the action decoder is 11056 and for the graph decoder is 7002. Hyperparameters were not tuned and were taken from other transformer-based text game works [1, 5]. Hyperparameter settings for ablations do not vary from the full Worldformer.

Encoders have an architecture similar to BERT [10] and decoders one similar to GPT-2 [27]—the rest of the hyperparameters are provided in Table 5.

| Hyperparameter type | Value |
|---|---|
| Text encoder | |
| Dictionary Tokenizer | Sentence piece |
| Num. layers | 6 |
| Num. attention heads | 6 |
| Feedforward network hidden size | 3072 |
| Input length | 1024 |
| Embedding size | 768 |
| Graph encoder | |
| Dictionary Tokenizer | Triple tokenizer |
| Num. layers | 6 |
| Num. attention heads | 6 |
| Feedforward network hidden size | 3072 |
| Input length | 1024 |
| Embedding size | 768 |
| Aggregator | |
| Num. layers | 2 |
| Num. attention heads | 2 |
| Feedforward network hidden size | 4096 |
| Input length | 2048 |
| Embedding size | 768 |
| Action Decoder | |
| Dictionary Tokenizer | White space tokenizer |
| Num. layers | 6 |
| Num. attention heads | 6 |
| Feedforward network hidden size | 3072 |
| Input length | 1024 |
| Embedding size | 768 |
| Graph Decoder | |
| Dictionary Tokenizer | Triple tokenizer |
| Num. layers | 6 |
| Num. attention heads | 6 |
| Feedforward network hidden size | 3072 |
| Input length | 1024 |
| Embedding size | 768 |
| Common | |
| Activation | gelu |
| Batch size | 16 |
| Dropout ratio | 0.1 |
| Gradient clip | 1.0 |
| Optimizer | Adam |
| Learning rate | $3 \times 10^{-4}$ |

Table 5: Hyperparameters used to train the Worldformer. It has a total of $\approx 380$ million trainable parameters. The triple tokenizer splits on individual parts of $\langle s, r, o \rangle$ as described in Section 4.2.

### A.3 Example Output Graphs and Actions

Here, we provide 3 examples of graphs and actions generated from the randomly drawn test example instances shown in JerichoWorld to provide a qualitative comparison across the different models.

```
Game: ludicorp
State:
    Location: Meeting Area
            A door to the south leads into the garden. A water cooler sits invitingly in the corner. More
                doors lead east and west. You can see a Coil of wire here.
    Observation: Dropped.
    Inventory: You are carrying:
                a Dragon Statue
                some Plant Pots
                a Long Ladder
```

```
                    a Gun
        Graph: ["Coil of wire", "in", "Meeting Area"],
            ["you", "have", "Plant Pots"],
            ["Water Cooler", "in", "Meeting Area"],
            ["you", "have", "Dragon Statue"],
            ["you", "have", "Long Ladder"],
            ["you", "in", "Meeting Area"],
            ["you", "have", "Gun"]
        Valid Actions: take wire, east, west, south, put dragon down, put pots down, put gun down, put ladder
            down
Act: take wire
Next State:
    Location: Meeting Area
            A door to the south leads into the garden. A water cooler sits invitingly in the corner. More
                doors lead east and west.
    Observation: Taken.
    Inventory: You are carrying:
                a Coil of wire
                a Dragon Statue
                some Plant Pots
                a Long Ladder
                a Gun
        Graph: ["you", "have", "Coil of wire"],
            ["you", "have", "Plant Pots"],
            ["Water Cooler", "in", "Meeting Area"],
            ["you", "have", "Dragon Statue"],
            ["you", "have", "Long Ladder"],
            ["you", "in", "Meeting Area"],
            ["you", "have", "Gun"]
        Valid Actions: put wire down, east, west, south, put dragon down, put pots down, put gun down, put
            ladder down
Predicted Next State Graphs:
    Rules: ["door to the", "in", "South"],
        ["more doors", "in", "east and west"]
        ["leads to the", "in", "garden"],
        ["you are carrying", "have", "some Plant Pots"],
        ["a water cooler sits", "in", "corner"],
        ["you are carrying", "have", "Dragon Statue"],
        ["you are carrying", "have", "a Long Ladder"],
        ["you", "in", "Meeting Area"],
        ["you are carrying", "have", "a Gun"]
    QA: ["you", "have", "Coil of wire"],
        ["door", "in", "South"],
        ["doors", "in", "east and west"]
        ["you", "have", "some Plant Pots"],
        ["water cooler", "in", "corner"],
        ["you", "have", "Dragon Statue"],
        ["you", "have", "a Long Ladder a Gun"],
        ["you", "in", "Meeting Area"]
    Seq2Seq: ["you", "have", "Pots"],
        ["you", "in", "Statue"],
        ["you", "have", "Ladder Gun"],
        ["you", "in", "Meeting Area"]
    GATA-W: ["you", "in", "Coil of wire"],
        ["you", "have", "Plant Pots"],
        ["Water Cooler", "in", "Meeting Area"],
        ["you", "have", "Dragon Statue"],
        ["you", "in", "Statue"],
        ["you", "have", "Long Ladder"],
        ["you", "in", "Meeting Area"],
        ["you", "have", "Gun"]
    Worldformer: ["you", "have", "Coil of wire"],
        ["you", "have", "Plant Pots"],
        ["Water Cooler", "in", "Meeting Area"],
        ["you", "in", "Dragon Statue"],
        ["you", "have", "Long Ladder"],
        ["you", "in", "Meeting Area"],
        ["you", "have", "Gun"]
Predicted Next State Valid Actions:
    Seq2Seq Actions: east, west, south, north, pick coil up, pick statue up, put ladder gun down, put
        meeting area down, put pots down
    CALM Actions: northwest, up, take plant, put plant in cooler, take water, take fish, south, wait, take
        all, north, take dragon, take cooler, southeast, take dragon statue, get all, east, west, take
        gun, northeast, southwest
    Worldformer Actions: east, west, south, north, put coil down, pick statue up, put ladder down, put pots
        down, put gun down

==================

Game: pentari
State:
```

```
Location: Armory
        Many death-dealing weapons of every type were stored here. Several tall racks probably held
            spears while shorter ones mounted against the wall stored various kinds of swords. Other
            wall mounts, also empty, give you no idea what sort of weapons may have been held by
            them. A large archway north is partially blocked by collapsed stones and rubble. You can
            see a jewel encrusted dagger here.
Observation: Armory
        Many death-dealing weapons of every type were stored here. Several tall racks probably held
                spears while shorter ones mounted against the wall stored various kinds of swords.
                Other wall mounts, also empty, give you no idea what sort of weapons may have been
                held by them. A large archway north is partially blocked by collapsed stones and
                rubble. You can see a jewel encrusted dagger here.
Inventory: You are carrying nothing.
Graph: ["jewel encrusted dagger", "in", "Armory" ],
    ["Armory", "west", "Main Hall"],
    ["you", "in", "Armory"]
Valid Actions: take other, east, north
Act: east
Next State:
Location: Main Hall
        This once majestic room was where visitors would come to relax and meet with the formal lord
            of the castle in a somewhat informal atmosphere. Several large comfortable couches are
            scattered about, dusty and altogether squalid. Many large tapestries still hang on the
            walls but are horribly faded from age. Large open archways lead east and west while a
            huge fireplace dominates the center of the room against the northern wall.
Observation: Main Hall
        This once majestic room was where visitors would come to relax and meet with the formal
                lord of the castle in a somewhat informal atmosphere. Several large comfortable
                couches are scattered about, dusty and altogether squalid. Many large tapestries still
                hang on the walls but are horribly faded from age. Large open archways lead east and
                west while a huge fireplace dominates the center of the room against the northern wall
                .
Inventory: You are carrying nothing.
Graph: ["couch", "in", "Main Hall"],
    ["jewel encrusted dagger", "in", "Armory" ],
    ["you", "in", "Main Hall"],
    ["Main Hall", "east", "Armory"],
    ["tapestry", "in","Main Hall"]
Valid Actions: east, west, south, north
Predicted Next State Graphs:
Rules: ["Several large comfortable couches", "in", "Main Hall"],
    ["many large tapestries", "in", "Main Hall"],
    ["many large tapestries", "is", "age"],
    ["huge fireplace dominates", "in", "center of room against northern wall"],
    ["death dealing weapons", "in", "Armory"],
    ["visitors", "in", "to relax"],
    ["archway north", "is", "blocked"],
    ["archway north", "in", "collapsed stones"],
    ["spears", "in", "several tall racks"],
    ["you", "in", "Armory"],
    ["you", "in", "see encrusted dagger"],
    ["you are carrying", "have", "nothing"],
    ["Main Hall", "east", "Armory"]
QA: ["couches", "in", "Main Hall"],
    ["tapestries", "in", "Main Hall"],
    ["tapestries", "is", "faded"],
    ["Main Hall", "east", "Armory"],
    ["fireplace", "in", "center of room"],
    ["weapons", "in", "Armory"],
    ["visitors", "in", "Armory"],
    ["spears", "is", "tall"],
    ["you", "in", "Armory"],
    ["you", "have", "nothing"]
Seq2Seq: ["couch", "in", "Main Hall"],
    ["jewel", "in", "Armory" ],
    ["you", "in", "Main Hall"],
    ["large", "in","Main Hall"]
GATA-W: ["couch", "in", "Main Hall"],
    ["jewel encrusted dagger", "in", "Armory" ],
    ["you", "in", "Main Hall"],
    ["tapestry", "in","Main Hall"]
Worldformer: ["couch", "in", "Main Hall"],
    ["dagger", "in", "Armory" ],
    ["you", "in", "Main Hall"],
    ["Main Hall", "east", "Armory"],
    ["faded", "in", "Main Hall"]
Predicted Next State Valid Actions:
Seq2Seq Actions: east, west, south, north, take jewel, take large, take armory, put couch down
CALM Actions: get dagger, northwest, out, up, down, exits, search tapestries, close door, south, search
        fireplace, enter fireplace, open fireplace, north, in, southeast, east, west, take dagger,
        northeast, southwest
```

```
    Worldformer Actions: east, west, south, north, take jewel, take large, examine large, examine couch

===================

Game: temple
State:
    Location: Dead End
            This part of the town is radically different from the parts closer to the tower. The roads are
                    narrower and the paving is irregular, sometimes stone slabs and sometimes cobblestones.
                    The buildings are tall but less well kept than before. There are still no windows or
                    doors, but there are a few overhead bridges from house to house. The road ends here and
                    the only way out is to the north. You can also see a wrought iron key and Charles
                    Bristow here.
    Observation: The cat jumps aside to avoid the projectile, but moves a bit too far. It falls down, but
            like most cats it escapes unhurt. The cat runs off to the north. The wrought iron key falls down
            again and hits one of the stone slabs. There is a hollow sound, much like there was some cavity
            below the slab. 'I used to have a cat, you know' Charles remarks.
    Inventory: You are carrying:
                    a vial labelled Mukhtar
                    the Caelestae Horriblis
                    two vials
                    a yellow paper
                    a hideous statue
    Graph: ["stone slab", "in", "Dead End"],
        ["slab", "is", "animate"],
        ["Charles' clothes", "in", "Charles Bristow"],
        ["clothes", "is", "animate"],
        ["clothes", "is", "equip"],
        ["you", "have", "mysterious vial"],
        ["you", "have", "vial labelled Mukhtar"],
        ["overhead bridge", "in", "Dead End"],
        ["you", "have", "Caelestae Horriblis"],
        ["you", "have", "yellow paper"],
        ["yellow paper", "is", "animate"],
        ["you", "have", "hideous statue"],
        ["hideous statue", "is", "animate"],
        ["sky", "in", "Dead End"],
        ["elliptcal building", "in", "Dead End"],
        ["you", "in", "Dead End"],
        ["entrance", "in", "Dead End"],
        ["Charles Bristow", "in", "Dead End"],
        ["wrought iron key", "in", "Dead End"]
    Valid Actions: take wrought, take paving, north, put mysterious down, put caelestae down, put paper
            down, put statue down, put mukhtar down, drop wrought against bridge
Act: put paper down
Next State:
    Location: Dead End
            This part of the town is radically different from the parts closer to the tower. The roads are
                    narrower and the paving is irregular, sometimes stone slabs and sometimes cobblestones.
                    The buildings are tall but less well kept than before. There are still no windows or
                    doors, but there are a few overhead bridges from house to house. The road ends here and
                    the only way out is to the north. You can also see a yellow paper, a wrought iron key
                    and Charles Bristow here.
    Observation: Dropped.
    Inventory: You are carrying:
                    a vial labelled Mukhtar
                    the Caelestae Horriblis
                    two vials
                    a hideous statue
    Graph: ["yellow paper", "in", "Dead End"],
        ["yellow paper", "is", "animate"],
        ["stone slab", "in", "Dead End"],
        ["stone slab", "is", "animate"],
        ["Charles' clothes", "in", "Charles Bristow"],
        ["clothes", "is", "animate"],
        ["clothes", "is", "equip"],
        ["you", "have", "mysterious vial"],
        ["you", "have", "vial labelled Mukhtar"],
        ["overhead bridge", "in", "Dead End"],
        ["you", "have", "Caelestae Horriblis"],
        ["you", "have", "hideous statue"],
        ["hideous statue", "is", "animate"],
        ["sky", "in", "Dead End"],
        ["elliptcal building", "in", "Dead End"],
        ["you", "in", "Dead End"],
        ["entrance", "in", "Dead End"],
        ["Charles Bristow", "in", "Dead End"],
        ["wrought iron key", "in", "Dead End"]
    Valid Actions: take wrought, take paving, north, put mysterious down, put caelestae down, take all, put
            statue down, put mukhtar down, drop wrought against bridge
Predicted Next State Graphs:
```

```
        Rules:["you", "in", "Dead End"],
           ["Dead End", "is", "radically different from the parts closer to the tower"],
           ["roads", "is", "narrower"],
           ["the paving", "is", "irregular"],
           ["the buildings", "is", "less well kept"],
           ["only way", "is", "north"],
           ["you can see", "in", "yellow paper"],
           ["you can see", "in", "a wrought iron key and Charles Bristow here"],
           ["you are carrying", "have", "a vial labelled Mukhtar"],
           ["you are carrying", "have", "the Caelestae Horriblis two vials"],
           ["you are carrying", "have", "a hideous statue"]
        QA: ["you", "in", "Dead End"],
           ["Dead End", "is", "animate"],
           ["road", "is", "animate"],
           ["paving", "is", "animate"],
           ["building", "is", "animate"],
           ["yellow paper", "in", "Dead End"],
           ["paper", "is", "animate"],
           ["key", "in", "Dead End"],
           ["key", "is", "animate"],
           ["Charles Bristow", "in", "Dead End"],
           ["Charles Bristow", "is", "animate"],
           ["you", "have", "a vial"],
           ["a vial", "is", "animate"],
           ["you", "have", "Caelestae Horriblis"],
           ["Caelestae Horriblis", "is", "animate"],
           ["you", "have", "two vials"],
           ["you", "have", "a hideous statue"]
           ["a hideous statue", "is", "animate"]
        Seq2Seq: ["you", "in", "Dead End"],
           ["Dead End", "is", "animate"],
           ["key", "in", "Dead End"],
           ["key", "is", "animate"],
           ["Charles Bristow", "in", "Dead End"],
           ["Charles Bristow", "is", "animate"],
           ["you", "have", "vial"],
           ["you", "have", "two vials"],
           ["you", "in", "statue"]
        GATA-W: ["yellow paper", "in", "Dead End"],
           ["stone slab", "in", "Dead End"],
           ["stone slab", "is", "animate"],
           ["Charles' clothes", "in", "Charles Bristow"],
           ["clothes", "is", "animate"],
           ["clothes", "is", "equip"],
           ["you", "have", "mysterious vial"],
           ["you", "have", "vial labelled Mukhtar"],
           ["overhead bridge", "in", "Dead End"],
           ["you", "in", "Caelestae Horriblis"],
           ["sky", "in", "Dead End"],
           ["elliptcal building", "in", "Dead End"],
           ["you", "in", "Dead End"],
           ["entrance", "in", "Dead End"],
           ["Charles Bristow", "in", "Dead End"],
           ["wrought iron key", "in", "Dead End"]
        Worldformer: ["yellow paper", "in", "Dead End"],
           ["yellow paper", "is", "animate"],
           ["stone slab", "in", "Dead End"],
           ["stone slab", "is", "animate"],
           ["Charles' clothes", "in", "Charles Bristow"],
           ["clothes", "is", "animate"],
           ["clothes", "is", "equip"],
           ["you", "have", "vial"],
           ["bridge", "in", "Dead End"],
           ["you", "have", "Caelestae Horriblis"],
           ["you", "have", "hideous statue"],
           ["hideous statue", "is", "animate"],
           ["sky", "in", "Dead End"],
           ["building", "in", "Dead End"],
           ["you", "in", "Dead End"],
           ["entrance", "in", "Dead End"],
           ["Charles Bristow", "in", "Dead End"],
           ["wrought iron key", "in", "Dead End"]
Predicted Next State Valid Actions:
     Seq2Seq Actions: east, west, south, north, drop vial, take key, take statue, take charles bristow
     CALM Actions: put paper down, drop vial, take key, up, down, put paper in cavity, put yellow paper down
           , take vial, drop key, drop all, open vial, south, get key, put yellow paper in cavity, take all,
           put vial in cavity, north, east, west, give vial to cat
     Worldformer Actions: south, east, west, north, take building, take entrance, drop vial, take charles
           bristow, take paper, drop vials, take sky, drop key, take statue, take bridge
```