# OpenReview forum: "Learning Knowledge Graph-based World Models of Textual Environments"
_NeurIPS.cc/2021/Conference — NeurIPS 2021 Poster_

### Official Review · Reviewer_QgLj · 2021-07-03

**Rating:** 7
**Confidence:** 5

**Summary:**

In this paper, the authors propose a novel system that tries to learn world model --- represented as knowledge graphs --- for agents playing text-based adventure games.

To learn the world model, the authors design two tasks:
- Knowledge graph prediction. More specifically, knowledge graph difference prediction. At a game step $t$, given the observation $O_t$, the set of valid actions $V_t$, the graph representation $G_t$, and an action $A \in V_t$, the model is required to predict the difference between $G_{t+1}$ and $G_t$, represented as a set of $<s, r, o>$ triplets. Using heuristics, the system deduces $G_{t} - G_{t+1}$ from $G_{t+1} - G_t$. Together, the set of graph difference (triplets) can be used to infer the full set of $G_{t+1}$, which represents the new game state $S_{t+1}$.
-Valid action prediction. Similarly, at a game step $t$, given $O_t$, $V_t$, $G_t$ and $A \in V_t$, the model is required to predict the set of valid actions $V_{t+1}$ at next game step.

Since both of the above tasks are essentially set (of sequences) generation tasks, the authors propose a modification on the standard autoregressive NLG paradigm. Specifically, the decoder outputs a sequence of elements where each element is a multi-word chunk (either a  $<s, r, o>$ triplet or an action), the author propose to remove the dependency across elements so that when generating a word, the decoder is only conditioned on the previous tokens within the same element. From Figure 3, the authors show that such chunks are concatenated with special tokens (e.g., [ACT], [TRIPLE]).

Another contribution is that the authors propose a multi-task architecture to jointly learn the aforementioned two tasks. Because the authors use triplets (sequences of tokens) to represent graphs, both the text encoder and graph encoder can be implemented as transformer-based encoders. The authors use an aggregator to combine the encodings from both encoders, denoted as $S_t$, which is later used to perform cross-attention with each of the individual encoding as the input to their corresponding decoder.



**Limitations And Societal Impact:**

Yes, the authors have discussed limitations and potential negative societal impact. I do not see a problem here.

**Main Review:**

Overall, this is a very interesting paper. The authors aim to learn a world model represents a model's understanding to the environment it is performing in. World model is a hot topic in model-based RL community, as cited by the authors, state prediction and action prediction seem to be a promising direction towards learning a "model".

The authors tackle the problem by using a static dataset, JerichoWorld, a set of trajectories along with transitions at every game step collected from a set of text-based games. I like this choice because 1) it circumvents extra difficulties in the interactive settings so that they can focus on the world model learning part, 2) but it can also be seen as an offline-RL setting since the static dataset can be treated as what an online agent collected and cached into the replay buffer.

I like how the authors formulate the world model as a graph --- there are prior works (such as Ammanabrolu and Riedl, 2019, Adhikari et al., 2020) suggest graph structures help text game agents as a memory and facilitate its planning. There is also a line of prior work focuses on representation learning in RL (e.g., Schwarzer et al., 2021), especially with contrastive predictive methods. This work in some sense brings the two lines of research together, explores how to learn a world model, in the format of a graph, by training with action and state prediction.

I agree with the authors that the valid action prediction and state prediction tasks are set generation tasks. The authors attempts to modify the standard teacher-forcing paradigm to make it work better on such set generation tasks.

The paper is clearly written, it is easy to follow.

I will put my questions and concerns in two parts:
## General (maybe highlevel) questions:

**(Q1)** Are the inputs to the world model sufficient? To answer this question, please first check if my understanding to the model input/output are correct (as listed in the summary review). As mentioned by the authors (L29-L30), text-based game environments are often partially observable. If that is the case, how does one predict $S_{t+1}$ from $O_t$, $V_t$, $G_t$ and $A$?

I can understand that within the current observation range, it is possible to predict the consequence of performing an action (e.g., take leaflet may cause the leaflet moving from mailbox to inventory). However, how does one predict something not in their observation range? For instance, in Figure 1, given all the information on left, plus the $A$ (go north), how to predict the right hand side information? How does the agent know what place is to the north, and what objects that place contains? I do expect a Seq2Seq model being able to memorize this from the training games, but I do not see how this can be generalized to unseen data points.

**(Q2)** If the inputs are somehow sufficient, are they all necessary? In the knowledge graph generation task proposed by the authors, the model learns: $O_t, V_t, G_t, A \rightarrow \delta$ where $\delta = G_{t+1} - G_t$. I fail to understand why $O_t$ and $V_t$ is necessary here. Doesn't $G_t$ contain all information in $O_t$? Is this redundant information by design? Similarly, in the valid action generation task, can a model learn to deduce $V_{t+1}$ from $G_{t+1}$ only? If so, I could imagine a reasoning path of $G_t, A \rightarrow \delta  \rightarrow G_{t+1}  \rightarrow V_{t+1}$.

## Detailed questions:

**(Q3)** What reason is behind the choice of representing graphs as sequences of triples, compared to the (probably) more straightforward practice of adjacency matrix and Graph Neural Nets? Presumably, the model should learn an permutation invariant representation of set. However, there might be ordering biases introduced by the current way of encoding the set of triples? Did the authors do special modifications to the positional embeddings in their Transformer-based encoders (e.g., reset positions at the first token of each triple/action)?

**(Q4)** How is $A$ integrated in the model? I fail to find it from Figure 3. With Figure 3 alone, the inputs to outputs mapping seems impossible.

**(Q5)** How does the proposed system learn to generalize to unseen objects and their affordances? Games in the Jericho collection can be quite different from each other, I wonder how do the proposed methods generalize to unseen and potentially very different games, where unseen objects have unseen affordances. It would be also nice to investigate deeper on the numbers shown in Table 1, what contributed to those numbers? What can be transferred across games and what not (i.e., what's an upperbound EM/F1 score for JerichoWorld)?

**(Q6)** I am curious to see a detailed analysis on the set of sequences (SOS) generation method. Ablation results in Table 2 and Table 3 suggest the SOS loss provide a non-trivial boost on performance, however, in experiments without SOS loss, do the authors sort the sub-sequences somehow? In Adhikari et al., they seem to suggest that sorting the sub-sequences are necessary in Seq2Seq learning. In addition, I am not intuitively convinced that cutting the dependencies between set members is the best way because there is arguably some kind of dependencies existing within a set (even to the least extent, set members should not be identical).

That said, what we want is essentially permutation invariant, rather than removing dependency. Given how the SOS loss is designed, a simple failure mode I can imagine is that the model may have hard time generating the first token in each of the triples/actions, because they all condition on the same special token [TRIPLE]/[ACT], as a result, does this lead to more duplicates during inference? I suggest the authors to provide more information to make this more convincing. Also, Ye et al., 2021 seems quite related, might worth checking.

**(Q7)** (minor point) It would be nice to see the learned world models are helpful in the interactive setting of text-based games rather than their static dataset version.

## Typos:
1. L47: that that.
2. L248: other other.
3. L260: Table 3 --> Table 2.

References:

1. Playing Text-Adventure Games with Graph-Based Deep Reinforcement Learning. Ammanabrolu and Riedl, NAACL 2019.
2. Learning Dynamic Belief Graphs to Generalize on Text-Based Games. Adhikari et al., NeurIPS 2020.
3. Data-Efficient Reinforcement Learning with Self-Predictive Representations. Schwarzer et al., ICLR 2021.
4. One2Set: Generating Diverse Keyphrases as a Set. Ye et al., ACL 2021.

**Time Spent Reviewing:**

6

---

> ### Author Response · Authors · 2021-08-10
> **Author Response**
>
> This is a very thorough review of our work and we thank the reviewer for their time and effort in producing it. We are encouraged by the comments and will attempt to clear up some of the concerns raised here.
>
> Q1. Are the inputs to the world model sufficient? To answer this question, please first check if my understanding to the model input/output are correct (as listed in the summary review). As mentioned by the authors (L29-L30), text-based game environments are often partially observable. …  how to predict the right hand side information? How does the agent know what place is to the north, and what objects that place contains? I do expect a Seq2Seq model being able to memorize this from the training games, but I do not see how this can be generalized to unseen data points.
> - The main idea in the JerichoWorld dataset is that there are high level patterns that are formed at a game level regarding adjacencies etc. For example, if you are west of a house and you go north, you get to the north side of a house or that kitchens are found inside houses. This kind of (genre-based) commonsense knowledge is what is really being tested by the dataset. The test games contain a relatively representative set of games in terms of genre to the training games and so we can reasonably expect a model regurgitating these high level patterns to perform reasonably well. The (non-trivial) zero shot performance of the models themselves indicates that these patterns exist and can be learned. A upper bound would likely only be possible when tested on non-expert humans (prior works on text game world generation [Fan et al. https://arxiv.org/abs/1911.09194, Urbanek et al. https://arxiv.org/abs/1903.03094, Ammanabrolu et al. https://arxiv.org/abs/2001.10161] indicate that (1) expert level humans achieve close to 100% performance especially on genres of games with less fantasy elements; and (2) performance varies for all humans based on how many fantasy elements are in the genre - fairy tales are harder to predict than detective games).
>
> Q2. If the inputs are somehow sufficient, are they all necessary?
> - Yes! We will make this point more clear in the text but O_t contains information that is not necessarily present in G_t. This point is noted in many previous text game works as well. The graphs as designed contain map or “declarative” information (Location X is west of Location Y, Character X is in Location Z, Object P is with Character X) but does not have procedural information (hitting a troll with a sword hurts it) found in the text or through character dialogue.
>
> Q3. What reason is behind the choice of representing graphs as sequences of triples, compared to the (probably) more straightforward practice of adjacency matrix and Graph Neural Nets?
> - We choose this representation for two reasons: (1) this lets us pre-train the transformer encoders with phrase level based masking; and (2) have a joint vocabulary across encoders enabling simultaneous learning of action <-> graph correspondences.
>
> Q4. How is A integrated in the model?
> - A is integrated by encoding with the bidirectional text encoder. We will fix Figure 3 to make this more clear.
>
> Q5. How does the proposed system learn to generalize to unseen objects and their affordances? ...What can be transferred across games and what not (i.e., what's an upperbound EM/F1 score for JerichoWorld)?
> - We refer the reviewer back to the answer for Q1
>
> Q6. I am curious to see a detailed analysis on the set of sequences (SOS) generation method. Ablation results in Table 2 and Table 3 suggest the SOS loss provide a non-trivial boost on performance, however, in experiments without SOS loss, do the authors sort the sub-sequences somehow?... I suggest the authors to provide more information to make this more convincing. Also, Ye et al., 2021 seems quite related, might worth checking.
> - The reviewer is correct in stating that cutting dependencies between set members is not necessarily the ideal way of generating these sets. At least as far as duplicates go, we post-hoc filter the set of generated triples/actions such that there are no duplicates (generating until we achieve such a set), there is no sorting done for the graph diff portions (though we do sort for the GATA-W baseline as that is based on Adhikari et al. and includes the additional <add>/<del> commands for the nodes). This is not unlike the CALM method in Yao et al. 2020 (though that does not use the set loss)  and we provide further comparison against that method and additional analysis to show the efficacy of the SOS loss. We thank the reviewer for the pointer to the contemporary Ye et al. 2021 reference, it is indeed very relevant to our work.

---

> > ### Comment · Reviewer_QgLj · 2021-08-25
> > **Response to the rebuttal**
> >
> > Thank for providing answers to my questions. The rebuttal has answered most of my questions. I will keep my score of 7.

---

### Official Review · Reviewer_cdHj · 2021-07-15

**Rating:** 7
**Confidence:** 3

**Summary:**

This paper presents a method for modelling state (via graph prediction) and action distributions in text adventure games. To do the former, this work proposes predicting incremental graph updates instead of entire graphs. For both, the authors propose predicting sets of sequences as opposed to sequence generation and show that this improves the accuracy of predicted action sets and environment state.

**Ethics Review Area:**

["I don’t know"]

**Limitations And Societal Impact:**

Yes.

**Main Review:**

This paper presents an original approach for modelling state and valid action spaces for text adventure games using pretrained language models. The work is clearly presented and well-motivated. My main concern has to do with the evaluation:

1. Is this estimation helpful for actual game-playing?
Given that the ultimate goal is to be more sample efficient due to having a good world model, I think it would be good to use this world-model to learn policies for these games, the the hypothesis being that using these models improve upon prior results. The numbers in Table 1 are all fairly low IMO and I am not confident that these results are actually helpful downstream.

2. I am unclear as to where the gains are coming from. Some ablations that I find missing include:
- QA & Seq2Seq with multitasking
- QA & Seq2Seq with set loss

3. What do the evaluation metrics mean? It is difficult to interpret F1 in the context of text adventure games. For example, what portion of the actions generated are actually parseable by the game parser?

Some questions for the authors:

- what are "lifted representations"? L38
- does this algorithm support removing nodes from the graph? some parts of the paper imply yes (L243) but the main exposition suggests no. L150
- is phrase-level masking actually better than normal LM masking? L165

**Time Spent Reviewing:**

4

---

> ### Author Response · Authors · 2021-08-10
> **Author Response**
>
> The reviewer's efforts towards improving our manuscript are much appreciated. We will address some concerns raised here.
>
> Q. Is this estimation helpful for actual game-playing? Given that the ultimate goal is to be more sample efficient due to having a good world model, I think it would be good to use this world-model to learn policies for these games, the the hypothesis being that using these models improve upon prior results. The numbers in Table 1 are all fairly low IMO and I am not confident that these results are actually helpful downstream.
> - We do not provide results for scores/reward when using this model over other baselines but other prior works (Ammanabrolu et al 2020, Yao et al 2020)  indicate that improved graph and action generation qualities respectively significantly increases scores received by agents in these environments. The Worldformer performs the functionality of both the Q*BERT agent which generates graphs (Ammanabrolu et al 2020) and the CALM agent which generates actions (Yao et al 2020), better than both of them. It can potentially be used as a drop-in replacement for any of these agents to perform reinforcement learning or can extend them to enable better planning via model based reinforcement learning. Even the relatively low numbers shown by both Q*BERT and CALM on our metrics translate to higher scores for downstream learned policies.
>
> Q. I am unclear as to where the gains are coming from. Some ablations that I find missing include: QA & Seq2Seq with multitasking, QA & Seq2Seq with set loss
> - QA, at least as presented in the Q*BERT baseline,  is incompatible with both multi-tasking and the set loss. Seq2Seq with multitasking and set loss are possible ablations and preliminary results indicate that both of these ablations significantly underperform compared to the full Worldformer. These ablations will be added in full in future revisions of this work. These results do not change the analysis provided in Lines 272-279, where we state that we believe that the majority of the gains come from the graph diff prediction component (as opposed to no diff) though there are smaller (but still significant) gains that come from multitask learning and SOS loss.
>
> Q. What do the evaluation metrics mean? It is difficult to interpret F1 in the context of text adventure games. For example, what portion of the actions generated are actually parseable by the game parser?
> - The reviewer is correct in stating that it is difficult to interpret F1 for actions. This is why we have provided the EM metric for this task. The EM metric provides the proportion of actions generated that can actually be parsed by the game parser, we will make this point more clear in the text.
>
> Q. what are "lifted representations"? L38
> - “Lifted representation” refer to the fact that the QA system is extractive and so is only capable of extracting or “lifting” phrases of text from the input context.
>
> Q. does this algorithm support removing nodes from the graph? some parts of the paper imply yes (L243) but the main exposition suggests no. L150
> - Yes it does, GATA-W (L243) explicitly encodes adds and removes in the sequence while the Worldformer only generates the nodes to add and then based on the assumptions made in Section 4.1, we are able to automatically create the set of nodes to remove every step. We will make this point more clear in future revisions.
>
> Q. is phrase-level masking actually better than normal LM masking? L165
> - Preliminary experiments for the graph generation components for this task and adjacent tasks such as (fantasy) story infilling (Donnahue et al. 2020 https://aclanthology.org/2020.acl-main.225/) indicate that phrase-level masking is better than normal LM masking.

---

> > ### Comment · Reviewer_cdHj · 2021-08-11
> > **Thanks**
> >
> > Thank you authors for the response. I have increased my score to 7. I would still like to see results for using this model over baselines during actual gameplay. I think this result would significantly strengthen this paper.

---

### Official Review · Reviewer_FHsa · 2021-07-17

**Rating:** 4
**Confidence:** 4

**Summary:**

This paper proposes a novel knowledge graph construction task based on attribute extraction from text game engines; and a set-of-sequence generation model that achieves good performance on the proposed tasks.

**Main Review:**

This paper proposes a novel knowledge graph construction task based on attribute extraction from text game engines. A set-of-sequence generation model, the Worldformer is proposed, which achieves good performance on the proposed tasks. I think this is a very clever way of creating knowledge graph construction tasks. And the two proposed tasks, on their own, are valid and interesting NLP tasks (though the human study might be needed, see my review at the end).

There are several major weaknesses this paper should addresses:

* Clarity

At first I did not find how the groundtruth graphs G_t and G_{t+1} are constructed. Then following the provided github repo I found this paper: https://openreview.net/pdf?id=Y1YtS9MZA75. Unfortunately, the dataset description paper still does not provide sufficient details. I guess the two papers are written by the same authors. If so, it would be good if more details about the dataset construction can be provided in the revised version. Specifically, how the tuples are extracted? Also why the average graph tuple number in a game is usually so small?

* The significance of the graphs prediction task

First, I did not see any theoretical or empirical justifications of the values of the proposed world model representation. Why this is the right world model for understanding the game texts? Is this world model better than simple sequence embedding of history? Why is it better than the model in [4]? The missing of the above answers makes the motivation of this paper not quite clear, as the addressed problem has not been well-verified yet according to the submission.

Second, the proposed model does not show a clear advantage over the Q*BERT baseline. The baseline formulates the graph construction task as question answering, therefore it is not surprising the baseline does not perform well on graph metrics but works well on token metrics. This raises another problem of the task significance -- the used intermediate metrics cannot clearly show which model is the best. It would make more sense if the graph construction models were used to enhance the game-playing agents, and the agents' performance was reported.

* Empirical comparison

Related to the previous bullet, the compared Q*BERT baseline was proposed to enhance a game-playing agent and performed quite well. As a fair comparison, the proposed Worldformer should be compared in the same setting.

The submission cited the related work [39], which was also proposed for valid action prediction. I am wondering why the work [39] was not compared in the experiments? Did I miss anything?

* A minor question:

Are the textual inputs guaranteed to be sufficient for the two tasks? I think conducting a human study would be necessary to show the quality of the tasks and their upperbound performance.

**Time Spent Reviewing:**

4

---

> ### Author Response · Authors · 2021-08-10
> **Author Response**
>
> We thank the reviewer for their time in providing this comprehensive review. We will attempt to address some of the concerns raised below.
>
> Q. At first I did not find how the groundtruth graphs G_t and G_{t+1} are constructed. Then following the provided github repo I found this paper: https://openreview.net/pdf?id=Y1YtS9MZA75. Unfortunately, the dataset description paper still does not provide sufficient details. I guess the two papers are written by the same authors. If so, it would be good if more details about the dataset construction can be provided in the revised version. Specifically, how the tuples are extracted? Also why the average graph tuple number in a game is usually so small?
> - The provided repo now contains the code for dataset construction and the link to the paper contains an (anonymized) datasheet containing all the details regarding dataset construction. A summary is that the data is collected by exploring by following walkthroughs of the games (in addition to extra random exploration) and extracting the graphs directly from the underlying game engine and converting object attributes and ids to human readable form. Text-based games are simulators with internal representations that capture significant and manipulable elements of the environment. For example, the game simulator knows the locations of things, the inventory of the player, whether doors are locked, etc. In the ground truth dataset construction paper, an agent exhaustively explores the games and accesses the internal variables of the simulator mapping these elements to <s, r, o> tuples such as <you, have, lantern>, and <bottle, in, kitchen>, and <bottle, is, animate> (“animate” being a special code used by the simulator to signify that the play can interact with the object), and <glasses, is, clothing> (“clothing” is a class of object that the simulator allows the verb “wear” to be applied to).
> - The average graph tuple length indicates the average number of new pieces of information that need to be added to the knowledge graph per state (the overall knowledge graph accumulates size at this rate) . This value varies dramatically based on the game, for some games such as snacktime with small numbers of characters/objects being as low as 2.33 but for other more complex games such as acorncourt as high as 36.14!!
>
> Q. First, I did not see any theoretical or empirical justifications of the values of the proposed world model representation. Why this is the right world model for understanding the game texts? Is this world model better than simple sequence embedding of history? Why is it better than the model in [4]? The missing of the above answers makes the motivation of this paper not quite clear, as the addressed problem has not been well-verified yet according to the submission.
> - Knowledge graphs have been repeatedly shown to be the right representation for understanding these texts in multiple prior works and are shown to be significantly better than simple sequence embeddings of history (Lines 33-55 in intro; lines 79-91 in our related work). We will make this point more explicit in future revisions. Given the robust literature that repeatedly demonstrates this, and given the trajectory of the literature that shows that improved knowledge graph construction and improved action prediction improves agent performance, we provide a multi-task training approach for the new benchmark task in https://openreview.net/pdf?id=Y1YtS9MZA75.
> - We provide a comprehensive series of ablations (Tables 2&3)  showing how the key components of our proposed world model (multitasking, SOS loss, graph diff prediction) all are critical for performance--performance deteriorates when removing those components. We further compare to many existing text game KG baselines for both of the tasks.
> - Lines 247-259 provide explicit analysis already on why our proposed model is better than [4] and we will make these points more clear. In particular, our evaluation shows that our multi-task approach is more accurate at the graph level (the most important metric because it captures graph structure), statistically indistinguishable from the token level, and archives this with less heuristic information ([4] uses fixed templates to extract graph elements).
>
> Q. The submission cited the related work [39], which was also proposed for valid action prediction. I am wondering why the work [39] was not compared in the experiments? Did I miss anything?
> - We have added these results now as part of Table 1 as well as additional analysis on this comparison (barring the embargo on revisions during rebuttals). We note that we significantly outperform [39] on the valid action prediction task as noted in the general comment.
>
> Q. Are the textual inputs guaranteed to be sufficient for the two tasks? I think conducting a human study would be necessary to show the quality of the tasks and their upperbound performance.
> - We apologize, we did not fully understand this question. Is the reviewer asking if all the information necessary to complete the task is provided in each textual input per state or per sequence? The sequence of actions needed to achieve rewards is another question entirely and usually requires multiple playthroughs of a game, for such a task (which we do not consider in this paper) - a human study would be quite helpful in determining upper-bound performance for such a task! Regarding such a human study, we would like to note that upper bound would likely only be approximated when tested on non-expert humans (prior works on text game world generation [Fan et al. https://arxiv.org/abs/1911.09194 and Urbanek et al. https://arxiv.org/abs/1903.03094, Ammanabrolu et al. https://arxiv.org/abs/2001.10161] indicate that (1) expert level humans achieve close to 100% performance especially on genres of games with less fantasy elements; and (2) performance varies for all humans based on how many fantasy elements are in the genre - fairy tales are harder to predict than detective games).
> - Further, we'd like to note that the scope of this paper is on whether the agent can accurately reconstruct the simulator’s ground-truth representation, which would not require a human evaluation due to the presence of the benchmark dataset.

---

### Official Review · Reviewer_swGy · 2021-07-26

**Rating:** 6
**Confidence:** 4

**Summary:**

The paper utilizes world models for text-based reinforcement learning agents to be aware of the change in the world based on one's own actions and what actions could be performed based on the current state of the world using Worldformer, a multi-task transformer for jointly learning to generate the knowledge graph differences based on the action taken and a set of valid action at the current state. Since the valid actions generated at the current step affect the action taken to change the world, these tasks are learned using the same loss function.

**Challenges**: Partial observability, combinatorial action space

**Limitations And Societal Impact:**

Yes, the broader statement explains the necessary details on the limitations and societal impacts.

**Main Review:**

**Major Comments:**
1. It would be helpful if the sample valid actions generated for some of the observations in the test set and the corresponding ground truth valid actions are shown side-by-side for comparison along with the graph additions and deletions. In addition to the numbers shown in Tables 2 and 3, qualitative examples could be very helpful in understanding the model.

2.  The results in the experiment section show how well the worldformer performs on the KG prediction and valid action prediction tasks for the held-out set. Does using this model for playing the games show any advantage over the other baselines in terms of scores/rewards? How will these learned world models be used for text-based games?

3. Does the checkmark in multitask mean the KG prediction and valid action task were learned independently?


**Minor Comments:**
1. Type in line 260 - table 2
2. Line 286 - 5 tokens

**Time Spent Reviewing:**

2

---

> ### Author Response · Authors · 2021-08-10
> **Author Response**
>
> We thank the reviewer for their thoughtful comments and are encouraged by them. Some concerns are addressed below.
>
> Q. It would be helpful if the sample valid actions generated for some of the observations in the test set and the corresponding ground truth valid actions are shown side-by-side for comparison along with the graph additions and deletions. In addition to the numbers shown in Tables 2 and 3, qualitative examples could be very helpful in understanding the model.
> - We have added multiple randomly selected examples showing sample generated graphs and actions for each of the baselines as well as the Worldformer in the Appendix. We cannot upload revisions during the rebuttal phase as per the rules of the conference but such changes have been made.
>
> Q. The results in the experiment section show how well the worldformer performs on the KG prediction and valid action prediction tasks for the held-out set. Does using this model for playing the games show any advantage over the other baselines in terms of scores/rewards? How will these learned world models be used for text-based games?
> - We do not provide results for scores/reward when using this model over other baselines but other prior works (Ammanabrolu et al 2020, Yao et al 2020)  indicate that improved graph and action generation qualities respectively significantly increases scores received by agents in these environments. The Worldformer performs the functionality of both the Q*BERT agent which generates graphs (Ammanabrolu et al 2020) and the CALM agent which generates actions (Yao et al 2020), better than both of them. It can potentially be used as a drop-in replacement for any of these agents to perform reinforcement learning or can extend them to enable better planning via model based reinforcement learning.
>
> Q. Does the checkmark in multitask mean the KG prediction and valid action task were learned independently?
> - A checkmark in multitask means that both tasks were learned jointly by optimizing the loss in Eq. 6. No checkmark in the column means that only the respective test task was trained for. We will make this point more clear.

---

### Author Response · Authors · 2021-08-10
**General Response to Authors Regarding Paper Changes**

We thank all the authors for their feedback. Based on the reviews, we have already incorporated some of the changes and wish to summarize them below. Note that these changes are in addition to the revisions promised in each of the individual author responses in future drafts.

1. We have added a comparison to the CALM method of valid action generation (Yao et al. 2020) as requested by Reviewer FHsa in Table 1 and note that its overall EM and F1 scores are respectively 13.79 and 19.11 (significantly lower than the Worldformer) - a full per game breakup will be provided in the revision. An analysis comparing this method to ours and on the additional utility of the SOS loss in enabling this has also been added (as requested by Reviewer QgLj).
2. Randomly selected qualitative examples of generated graphs and actions for all of the baselines and the Worldformer have been added to the appendix as requested by Reviewer swGy. We are posting these in a separate comment in response to this one as we are not allowed to update the paper/appendix at this point.
3. We have reiterated the scope of the paper in the introduction to whether the agent can accurately reconstruct the simulator’s ground-truth representation based on the benchmark dataset and relations to human evaluations/upperbound performance.
4. All of the minor typos that the reviewers have caught have been fixed.

---

> ### Author Response · Authors · 2021-08-10
> **Randomly Selected Qualitative Example 1 of KGs and Actions Across Baselines and the Worldformer**
>
> ```
> Game: ludicorp
> State:
>     Location: Meeting Area
>               A door to the south leads into the garden. A water cooler sits invitingly in the corner. More doors lead east and west. You can see a Coil of wire here.
>     Observation: Dropped.
>     Inventory: You are carrying:
>                   a Dragon Statue
>                   some Plant Pots
>                   a Long Ladder
>                   a Gun
>     Graph: ["Coil of wire", "in", "Meeting Area"],
>         ["you", "have", "Plant Pots"],
>         ["Water Cooler", "in", "Meeting Area"],
>         ["you", "have", "Dragon Statue"],
>         ["you", "have", "Long Ladder"],
>         ["you", "in", "Meeting Area"],
>         ["you", "have", "Gun"]
>     Valid Actions: take wire, east, west, south, put dragon down, put pots down, put gun down, put ladder down
> Act: take wire
> Next State:
>     Location: Meeting Area
>               A door to the south leads into the garden. A water cooler sits invitingly in the corner. More doors lead east and west.
>     Observation: Taken.
>     Inventory: You are carrying:
>                   a Coil of wire
>                   a Dragon Statue
>                   some Plant Pots
>                   a Long Ladder
>                   a Gun
>     Graph: ["you", "have", "Coil of wire"],
>         ["you", "have", "Plant Pots"],
>         ["Water Cooler", "in", "Meeting Area"],
>         ["you", "have", "Dragon Statue"],
>         ["you", "have", "Long Ladder"],
>         ["you", "in", "Meeting Area"],
>         ["you", "have", "Gun"]
>     Valid Actions: put wire down, east, west, south, put dragon down, put pots down, put gun down, put ladder down
> Predicted Next State Graphs:
>     Rules: ["door to the", "in", "South"],
>         ["more doors", "in", "east and west"]
>         ["leads to the", "in", "garden"],
>         ["you are carrying", "have", "some Plant Pots"],
>         ["a water cooler sits", "in", "corner"],
>         ["you are carrying", "have", "Dragon Statue"],
>         ["you are carrying", "have", "a Long Ladder"],
>         ["you", "in", "Meeting Area"],
>         ["you are carrying", "have", "a Gun"]
>     QA: ["you", "have", "Coil of wire"],
>         ["door", "in", "South"],
>         ["doors", "in", "east and west"]
>         ["you", "have", "some Plant Pots"],
>         ["water cooler", "in", "corner"],
>         ["you", "have", "Dragon Statue"],
>         ["you", "have", "a Long Ladder a Gun"],
>         ["you", "in", "Meeting Area"]
>     Seq2Seq: ["you", "have", "Pots"],
>         ["you", "in", "Statue"],
>         ["you", "have", "Ladder Gun"],
>         ["you", "in", "Meeting Area"]
>     GATA-W: ["you", "in", "Coil of wire"],
>         ["you", "have", "Plant Pots"],
>         ["Water Cooler", "in", "Meeting Area"],
>         ["you", "have", "Dragon Statue"],
>         ["you", "in", "Statue"],
>         ["you", "have", "Long Ladder"],
>         ["you", "in", "Meeting Area"],
>         ["you", "have", "Gun"]
>     Worldformer: ["you", "have", "Coil of wire"],
>         ["you", "have", "Plant Pots"],
>         ["Water Cooler", "in", "Meeting Area"],
>         ["you", "in", "Dragon Statue"],
>         ["you", "have", "Long Ladder"],
>         ["you", "in", "Meeting Area"],
>         ["you", "have", "Gun"]
> Predicted Next State Valid Actions:
>     Seq2Seq Actions: east, west, south, north, pick coil up, pick statue up, put ladder gun down, put meeting area down, put pots down
>     CALM Actions: northwest, up, take plant, put plant in cooler, take water, take fish, south, wait, take all, north, take dragon, take cooler, southeast, take dragon statue, get all, east, west, take gun, northeast, southwest
>     Worldformer Actions: east, west, south, north, put coil down, pick statue up, put ladder down, put pots down, put gun down
> ```

---

> ### Author Response · Authors · 2021-08-10
> **Randomly Selected Qualitative Example 2 of KGs and Actions Across Baselines and the Worldformer**
>
> ```
> Game: pentari
> State:
>     Location: Armory
>               Many death-dealing weapons of every type were stored here.  Several tall racks probably held spears while shorter ones mounted against the wall stored various kinds of swords.  Other wall mounts, also empty, give you no idea what sort of weapons may have been held by them.  A large archway north is partially blocked by collapsed stones and rubble. You can see a jewel encrusted dagger here.
>     Observation: Armory
>                  Many death-dealing weapons of every type were stored here.  Several tall racks probably held spears while shorter ones mounted against the wall stored various kinds of swords.  Other wall mounts, also empty, give you no idea what sort of weapons may have been held by them.  A large archway north is partially blocked by collapsed stones and rubble. You can see a jewel encrusted dagger here.
>     Inventory: You are carrying nothing.
>     Graph: ["jewel encrusted dagger", "in", "Armory" ],
>         ["Armory", "west", "Main Hall"],
>         ["you", "in", "Armory"]
>     Valid Actions: take other, east, north
> Act: east
> Next State:
>     Location: Main Hall
>               This once majestic room was where visitors would come to relax and meet with the formal lord of the castle in a somewhat informal atmosphere. Several large comfortable couches are scattered about, dusty and altogether squalid.  Many large tapestries still hang on the walls but are horribly faded from age.  Large open archways lead east and west while a huge fireplace dominates the center of the room against the northern wall.
>     Observation: Main Hall
>                  This once majestic room was where visitors would come to relax and meet with the formal lord of the castle in a somewhat informal atmosphere. Several large comfortable couches are scattered about, dusty and altogether squalid.  Many large tapestries still hang on the walls but are horribly faded from age.  Large open archways lead east and west while a huge fireplace dominates the center of the room against the northern wall.
>     Inventory: You are carrying nothing.
>     Graph: ["couch", "in", "Main Hall"],
>         ["jewel encrusted dagger", "in", "Armory" ],
>         ["you", "in", "Main Hall"],
>         ["Main Hall", "east", "Armory"],
>         ["tapestry", "in","Main Hall"]
>     Valid Actions: east, west, south, north
> Predicted Next State Graphs:
>     Rules: ["Several large comfortable couches", "in", "Main Hall"],
>         ["many large tapestries", "in", "Main Hall"],
>         ["many large tapestries", "is", "age"],
>         ["huge fireplace dominates", "in", "center of room against northern wall"],
>         ["death dealing weapons", "in", "Armory"],
>         ["visitors", "in", "to relax"],
>         ["archway north", "is", "blocked"],
>         ["archway north", "in", "collapsed stones"],
>         ["spears", "in", "several tall racks"],
>         ["you", "in", "Armory"],
>         ["you", "in", "see encrusted dagger"],
>         ["you are carrying", "have", "nothing"],
>         ["Main Hall", "east", "Armory"]
>     QA: ["couches", "in", "Main Hall"],
>         ["tapestries", "in", "Main Hall"],
>         ["tapestries", "is", "faded"],
>         ["Main Hall", "east", "Armory"],
>         ["fireplace", "in", "center of room"],
>         ["weapons", "in", "Armory"],
>         ["visitors", "in", "Armory"],
>         ["spears", "is", "tall"],
>         ["you", "in", "Armory"],
>         ["you", "have", "nothing"]
>     Seq2Seq: ["couch", "in", "Main Hall"],
>         ["jewel", "in", "Armory" ],
>         ["you", "in", "Main Hall"],
>         ["large", "in","Main Hall"]
>     GATA-W: ["couch", "in", "Main Hall"],
>         ["jewel encrusted dagger", "in", "Armory" ],
>         ["you", "in", "Main Hall"],
>         ["tapestry", "in","Main Hall"]
>     Worldformer: ["couch", "in", "Main Hall"],
>         ["dagger", "in", "Armory" ],
>         ["you", "in", "Main Hall"],
>         ["Main Hall", "east", "Armory"],
>         ["faded", "in", "Main Hall"]
> Predicted Next State Valid Actions:
>     Seq2Seq Actions: east, west, south, north, take jewel, take large, take armory, put couch down
>     CALM Actions: get dagger, northwest, out, up, down, exits, search tapestries, close door, south, search fireplace, enter fireplace, open fireplace, north, in, southeast, east, west, take dagger, northeast, southwest
>     Worldformer Actions: east, west, south, north, take jewel, take large, examine large, examine couch
>
> ```

---

> ### Author Response · Authors · 2021-08-10
> **Randomly Selected Qualitative Example 3 of KGs and Actions Across Baselines and the Worldformer**
>
> ```
> Game: temple
> State:
>     Location: Dead End
>               This part of the town is radically different from the parts closer to the tower. The roads are narrower and the paving is irregular, sometimes stone slabs and sometimes cobblestones. The buildings are tall but less well kept than before. There are still no windows or doors, but there are a few overhead bridges from house to house. The road ends here and the only way out is to the north. You can also see a wrought iron key and Charles Bristow here.
>     Observation: The cat jumps aside to avoid the projectile, but moves a bit too far. It falls down, but like most cats it escapes unhurt. The cat runs off to the north. The wrought iron key falls down again and hits one of the stone slabs. There is a hollow sound, much like there was some cavity below the slab. 'I used to have a cat, you know' Charles remarks.
>     Inventory: You are carrying:
>                 a vial labelled Mukhtar
>                 the Caelestae Horriblis
>                 two vials
>                 a yellow paper
>                 a hideous statue
>     Graph: ["stone slab",  "in", "Dead End"],
>         ["slab", "is", "animate"],
>         ["Charles' clothes", "in", "Charles Bristow"],
>         ["clothes", "is", "animate"],
>         ["clothes", "is", "equip"],
>         ["you", "have", "mysterious vial"],
>         ["you", "have", "vial labelled Mukhtar"],
>         ["overhead bridge", "in", "Dead End"],
>         ["you", "have", "Caelestae Horriblis"],
>         ["you", "have", "yellow paper"],
>         ["yellow paper", "is", "animate"],
>         ["you", "have", "hideous statue"],
>         ["hideous statue", "is", "animate"],
>         ["sky", "in", "Dead End"],
>         ["elliptcal building", "in", "Dead End"],
>         ["you", "in", "Dead End"],
>         ["entrance", "in", "Dead End"],
>         ["Charles Bristow", "in", "Dead End"],
>         ["wrought iron key", "in", "Dead End"]
>     Valid Actions: take wrought, take paving, north, put mysterious down, put caelestae down, put paper down, put statue down, put mukhtar down, drop wrought against bridge
> Act: put paper down
> Next State:
>     Location: Dead End
>               This part of the town is radically different from the parts closer to the tower. The roads are narrower and the paving is irregular, sometimes stone slabs and sometimes cobblestones. The buildings are tall but less well kept than before. There are still no windows or doors, but there are a few overhead bridges from house to house. The road ends here and the only way out is to the north. You can also see a yellow paper, a wrought iron key and Charles Bristow here.
>     Observation: Dropped.
>     Inventory: You are carrying:
>                 a vial labelled Mukhtar
>                 the Caelestae Horriblis
>                 two vials
>                 a hideous statue
>     Graph: ["yellow paper",  "in", "Dead End"],
>         ["yellow paper", "is", "animate"],
>         ["stone slab",  "in", "Dead End"],
>         ["stone slab", "is", "animate"],
>         ["Charles' clothes", "in", "Charles Bristow"],
>         ["clothes", "is", "animate"],
>         ["clothes", "is", "equip"],
>         ["you", "have", "mysterious vial"],
>         ["you", "have", "vial labelled Mukhtar"],
>         ["overhead bridge", "in", "Dead End"],
>         ["you", "have", "Caelestae Horriblis"],
>         ["you", "have", "hideous statue"],
>         ["hideous statue", "is", "animate"],
>         ["sky", "in", "Dead End"],
>         ["elliptcal building", "in", "Dead End"],
>         ["you", "in", "Dead End"],
>         ["entrance", "in", "Dead End"],
>         ["Charles Bristow", "in", "Dead End"],
>         ["wrought iron key", "in", "Dead End"]
>     Valid Actions: take wrought, take paving, north, put mysterious down, put caelestae down, take all, put statue down, put mukhtar down, drop wrought against bridge
> Predicted Next State Graphs:
>     Rules:["you", "in", "Dead End"],
>         ["Dead End", "is", "radically different from the parts closer to the tower"],
>         ["roads", "is", "narrower"],
>         ["the paving", "is", "irregular"],
>         ["the buildings", "is", "less well kept"],
>         ["only way", "is", "north"],
>         ["you can see", "in", "yellow paper"],
>         ["you can see", "in", "a wrought iron key and Charles Bristow here"],
>         ["you are carrying", "have", "a vial labelled Mukhtar"],
>         ["you are carrying", "have", "the Caelestae Horriblis two vials"],
>         ["you are carrying", "have", "a hideous statue"]
>     QA: ["you", "in", "Dead End"],
>         ["Dead End", "is", "animate"],
>         ["road", "is", "animate"],
>         ["paving", "is", "animate"],
>         ["building", "is", "animate"],
>         ["yellow paper", "in", "Dead End"],
>         ["paper", "is", "animate"],
>         ["key", "in", "Dead End"],
>         ["key", "is", "animate"],
>         ["Charles Bristow", "in", "Dead End"],
>         ["Charles Bristow", "is", "animate"],
>         ["you", "have", "a vial"],
>         ["a vial", "is", "animate"],
>         ["you", "have", "Caelestae Horriblis"],
>         ["Caelestae Horriblis", "is", "animate"],
>         ["you", "have", "two vials"],
>         ["you", "have", "a hideous statue"]
>         ["a hideous statue", "is", "animate"]
>     Seq2Seq: ["you", "in", "Dead End"],
>         ["Dead End", "is", "animate"],
>         ["key", "in", "Dead End"],
>         ["key", "is", "animate"],
>         ["Charles Bristow", "in", "Dead End"],
>         ["Charles Bristow", "is", "animate"],
>         ["you", "have", "vial"],
>         ["you", "have", "two vials"],
>         ["you", "in", "statue"]
>     GATA-W: ["yellow paper",  "in", "Dead End"],
>         ["stone slab",  "in", "Dead End"],
>         ["stone slab", "is", "animate"],
>         ["Charles' clothes", "in", "Charles Bristow"],
>         ["clothes", "is", "animate"],
>         ["clothes", "is", "equip"],
>         ["you", "have", "mysterious vial"],
>         ["you", "have", "vial labelled Mukhtar"],
>         ["overhead bridge", "in", "Dead End"],
>         ["you", "in", "Caelestae Horriblis"],
>         ["sky", "in", "Dead End"],
>         ["elliptcal building", "in", "Dead End"],
>         ["you", "in", "Dead End"],
>         ["entrance", "in", "Dead End"],
>         ["Charles Bristow", "in", "Dead End"],
>         ["wrought iron key", "in", "Dead End"]
>     Worldformer: ["yellow paper",  "in", "Dead End"],
>         ["yellow paper", "is", "animate"],
>         ["stone slab",  "in", "Dead End"],
>         ["stone slab", "is", "animate"],
>         ["Charles' clothes", "in", "Charles Bristow"],
>         ["clothes", "is", "animate"],
>         ["clothes", "is", "equip"],
>         ["you", "have", "vial"],
>         ["bridge", "in", "Dead End"],
>         ["you", "have", "Caelestae Horriblis"],
>         ["you", "have", "hideous statue"],
>         ["hideous statue", "is", "animate"],
>         ["sky", "in", "Dead End"],
>         ["building", "in", "Dead End"],
>         ["you", "in", "Dead End"],
>         ["entrance", "in", "Dead End"],
>         ["Charles Bristow", "in", "Dead End"],
>         ["wrought iron key", "in", "Dead End"]
> Predicted Next State Valid Actions:
>     Seq2Seq Actions: east, west, south, north, drop vial, take key, take statue, take charles bristow
>     CALM Actions: put paper down, drop vial, take key, up, down, put paper in cavity, put yellow paper down, take vial, drop key, drop all, open vial, south, get key, put yellow paper in cavity, take all, put vial in cavity, north, east, west, give vial to cat
>     Worldformer Actions: south, east, west, north, take building, take entrance, drop vial, take charles bristow, take paper, drop vials, take sky, drop key, take statue, take bridge
> ```

---

### Decision · Program_Chairs · 2021-09-27

**Decision:**

Accept (Poster)

**Comment:**

This paper jointly learns a policy (action distribution) and world model (updates to a graph representation of the environment) in a text adventure. While interesting, I'm not seeing the obvious application of this (despite having read the ethics impact section of the paper), and have some concerns about text adventure games as a domain — however the meta-review stage of the review process is not an appropriate point for me to bring these concerns to the authors' attention, so I will happily discount them and focus on the reviews and discussion.

The reviewer consensus leans towards acceptance, after some discussion. I would have liked to see a follow-up response from reviewer FHsa to the author rebuttal, and have no evidence that they have read it or considered updating their score despite my prompts, which is a bit disappointing. Less crucially, it would have been nice to get an author response to the follow-on comments by reviewer cdHj, but this is not as important since that reviewer updated their score to recommend acceptance.

Ultimately, when discounting the score of reviewer FHsa, the median and mean scores are in acceptance territory. I confess I do not fully understand reviewer FHsa's argument against the paper, and felt the author response was detailed. In the absence of this reviewer's willingness to defend their appraisal, I recommend acceptance.